# Highly multiplexed immunofluorescence imaging of human tissues and tumors using t-CyCIF and conventional optical microscopes

Jia-Ren Lin[1,2†], Benjamin Izar[1,2,3,4†], Shu Wang[1,5], Clarence Yapp[1], Shaolin Mei[1,3], Parin M Shah[3], Sandro Santagata[1,2,6,7], Peter K Sorger[1,2]*

[1]Laboratory of Systems Pharmacology, Harvard Medical School, Boston, United States; [2]Ludwig Center for Cancer Research at Harvard, Harvard Medical School, Boston, United States; [3]Department of Medical Oncology, Dana-Farber Cancer Institute, Boston, United States; [4]Broad Institute of MIT and Harvard, Cambridge, United States; [5]Harvard Graduate Program in Biophysics, Harvard University, Cambridge, United States; [6]Department of Pathology, Brigham and Women's Hospital, Harvard Medical School, Boston, United States; [7]Department of Oncologic Pathology, Dana-Farber Cancer Institute, Boston, United States

**Abstract** The architecture of normal and diseased tissues strongly influences the development and progression of disease as well as responsiveness and resistance to therapy. We describe a tissue-based cyclic immunofluorescence (t-CyCIF) method for highly multiplexed immuno-fluorescence imaging of formalin-fixed, paraffin-embedded (FFPE) specimens mounted on glass slides, the most widely used specimens for histopathological diagnosis of cancer and other diseases. t-CyCIF generates up to 60-plex images using an iterative process (a cycle) in which conventional low-plex fluorescence images are repeatedly collected from the same sample and then assembled into a high-dimensional representation. t-CyCIF requires no specialized instruments or reagents and is compatible with super-resolution imaging; we demonstrate its application to quantifying signal transduction cascades, tumor antigens and immune markers in diverse tissues and tumors. The simplicity and adaptability of t-CyCIF makes it an effective method for pre-clinical and clinical research and a natural complement to single-cell genomics.
DOI: https://doi.org/10.7554/eLife.31657.001

*For correspondence:
peter_sorger@hms.harvard.edu

†These authors contributed equally to this work

## Introduction

Histopathology is among the most important and widely used methods for diagnosing human disease and studying the development of multicellular organisms. As commonly performed, imaging of formalin-fixed, paraffin-embedded (FFPE) tissue has relatively low dimensionality, primarily comprising Hematoxylin and Eosin (H&E) staining supplemented by immunohistochemistry (IHC). The potential of IHC to aid in diagnosis and prioritization of therapy is well established (*Bodenmiller, 2016*), but IHC is primarily a single-channel method: imaging multiple antigens usually involves the analysis of sequential tissue slices or harsh stripping protocols (although limited multiplexing is possible using IHC and bright-field imaging [*Stack et al., 2014*; *Tsujikawa et al., 2017*]). Antibody detection via formation of a brown diamino-benzidine (DAB) or similar precipitates are also less quantitative than fluorescence (*Rimm, 2006*). The limitations of IHC are particularly acute when it is necessary to quantify complex cellular states and multiple cell types, such as tumor infiltrating regulatory and cytotoxic T cells (*Postow et al., 2015*) in parallel with tissue and pharmaco-dynamic markers.

**eLife digest** To diagnose a disease such as cancer, doctors sometimes take small tissue samples called biopsies from the affected area. These biopsies are then thinly sliced and treated with dyes to identify healthy and cancerous cells. However, clinicians and scientists often need to look into what happens inside individual cells in the tissues so they can understand how cancers arise and progress. This helps them to identify different types of tumor cells and to tailor the best treatment for the patient.

To do so, a number of proteins (the molecules involved in nearly all life's processes) need to be tracked in healthy and diseased cells and tissues. This can be done thanks to a range of methods known as immunofluorescence microscopy, but following different proteins on the same slice of a sample is difficult. However, a new type of immunofluorescence known as t-CyCIF may be a solution.

With this technique, a fluorescent compound is applied that will bind to a specific protein of interest. A microscope can pick up the light from the compound when the sample is imaged, which reveals the protein's location in the cell or tissue. Then, a substance is used that deactivates the fluorescence signal. After this, another compound that binds to a new type of protein is used, and imaged. This cycle is repeated several times to locate different proteins. Lastly, the individual images are processed and stitched together to reveal the cells and their internal structures.

Here, Lin, Izar et al. showed that t-CyCIF could be used to study biopsies and to obtain images that covered a large area of healthy human tissues and tumors. The technique helped to track over 60 different proteins in normal and tumor tissue samples from human patients. Several sets of experiments showed that t-CyCIF could uncover the molecular mechanisms that are disrupted during cancer, but also reveal the complexity of a single tumor. In fact, as shown with biopsies of brain cancer, cancerous cells in a tumor can be strikingly different, even when they are close to each other. Finally, the method helped to pinpoint which types of immune cells are involved in fighting a kidney tumor. Overall, such information cannot be obtained with conventional methods, yet is crucial for diagnosis and treatment.

Most laboratories can readily use t-CyCIF since the technique is open source and requires equipment that is easily accessible. In fact, the technique should soon be used to assess how well certain drugs help the immune system combat cancer. Ultimately, better use of biopsies is key to customizing cancer care.

DOI: https://doi.org/10.7554/eLife.31657.002

Advances in DNA and RNA profiling have dramatically improved our understanding of oncogenesis and propelled the development of targeted anticancer drugs (*Garraway and Lander, 2013*). Sequence data are particularly useful when an oncogenic driver is both a drug target and a biomarker of drug response, such as $BRAF^{V600E}$ in melanoma (*Chapman et al., 2011*) or *BCR-ABL* in chronic myelogenous leukemia (*Druker and Lydon, 2000*). However, in the case of drugs that act through cell non-autonomous mechanisms, such as immune checkpoint inhibitors, tumor-drug interaction must be studied in the context of multicellular environments that include both cancer and non-malignant stromal and infiltrating immune cells. Multiple studies have established that these components of the tumor microenvironment strongly influence the initiation, progression and metastasis of cancer (*Hanahan and Weinberg, 2011*) and the magnitude of responsiveness or resistance to immunotherapies (*Tumeh et al., 2014*).

Single-cell transcriptome profiling provides a means to dissect tumor ecosystems at a molecular level and quantify cell types and states (*Tirosh et al., 2016*). However, single-cell sequencing usually requires disaggregation of tissues, resulting in loss of spatial context (*Tirosh et al., 2016*; *Patel et al., 2014*). As a consequence, a variety of multiplexed approaches to analyzing tissues have recently been developed with the goal of simultaneously assaying cell identity, state, and morphology (*Giesen et al., 2014*; *Gerdes et al., 2013*; *Micheva and Smith, 2007*; *Remark et al., 2016*; *Gerner et al., 2012*). For example, FISSEQ (*Lee et al., 2014*) enables genome-scale RNA profiling of tissues at single-cell resolution, and multiplexed ion beam imaging (MIBI) and imaging mass cytometry achieve a high degree of multiplexing using antibodies as reagents, metals as labels and

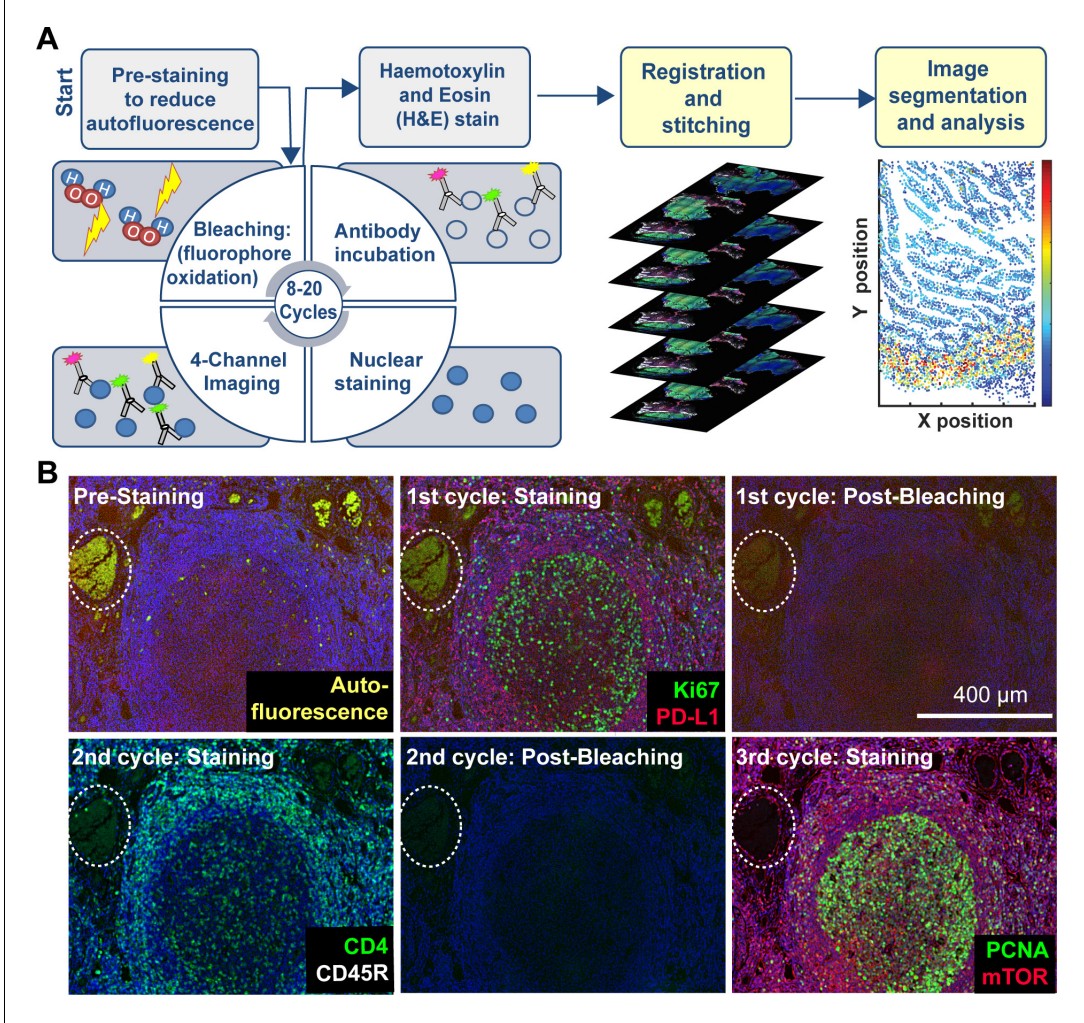

**Figure 1.** Steps in the t-CyCIF process. (**A**) Schematic of the cyclic process whereby t-CyCIF images are assembled via multiple rounds of four-color imaging. (**B**) Image of human tonsil prior to pre-staining and then over the course of three rounds of t-CyCIF. The dashed circle highlights a region with auto-fluorescence in both green and red channels (used for Alexa-488 and Alexa-647, respectively) and corresponds to a strong background signal. With subsequent inactivation and staining cycles (three cycles shown here), this background signal becomes progressively less intense; the phenomenon of decreasing background signal and increasing signal-to-noise ratio as cycle number increases was observed in several staining settings (see also *Figure 1—figure supplement 1*).

DOI: https://doi.org/10.7554/eLife.31657.003

The following figure supplements are available for figure 1:

**Figure supplement 1.** Reduction in background signal intensity with repeated cycles of bleaching.
DOI: https://doi.org/10.7554/eLife.31657.004

**Figure supplement 2.** t-CyCIF using antibodies labelled with Zenon Alexa-555 Fab fragments.
DOI: https://doi.org/10.7554/eLife.31657.005

mass spectrometry as a detection modality (*Giesen et al., 2014*; *Angelo et al., 2014*). Despite the potential of these new methods, they require specialized instrumentation and consumables, which is one reason that the great majority of basic and clinical studies still rely on H&E and single-channel IHC staining. Moreover, methods that involve laser ablation of samples such as MIBI inherently have a lower resolution than optical imaging.

Thus, there remains a need for highly multiplexed tissue analysis methods that (i) minimize the requirement for specialized instruments and costly, proprietary reagents, (ii) work with conventionally prepared FFPE tissue specimens collected in clinical practice and research settings, (iii) enable imaging of ca. 50 antigens at subcellular resolution across a wide range of cell and tumor types, (iv)

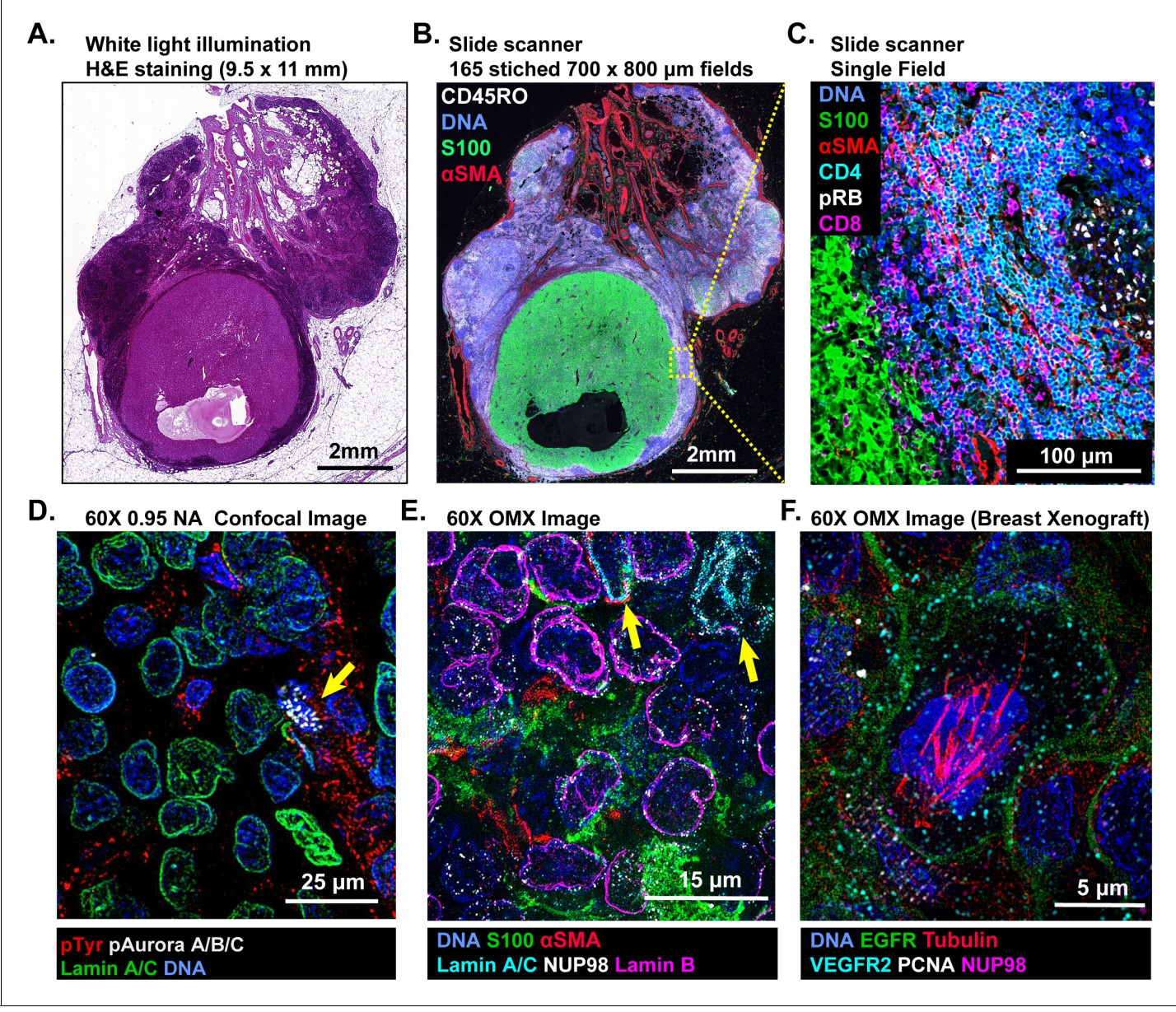

**Figure 2.** Multi-scale imaging of t-CyCIF specimens. (**A**) Bright-field H&E image of a metastasectomy specimen that includes a large metastatic melanoma lesion and adjacent benign tissue. The H&E staining was performed after the same specimen had undergone t-CyCIF. (**B**) Representative t-CyCIF staining of the specimen shown in (**A**) stitched together using the *Ashlar* software from 165 successive CyteFinder fields using a 20X/0.8NA objective. (**C**) One field from (**B**) at the tumor-normal junction demonstrating staining for S100-postive malignant cells, α-SMA positive stroma, T lymphocytes (positive for CD3, CD4 and CD8), and the proliferation marker phospho-RB (pRB). (**D**) A melanoma tumor imaged on a GE INCell Analyzer 6000 confocal microscope to demonstrate sub-cellular and sub-organelle structures. This specimen was stained with phospho-Tyrosine (pTyr), Lamin A/C and p-Aurora A/B/C and imaged with a 60X/0.95NA objective. pTyr is localized in membrane in patches associated with receptor-tyrosine kinase, visible here as red punctate structures. Lamin A/C is a nuclear membrane protein that outlines the vicinity of the cell nucleus in this image. Aurora kinases A/B/C coordinate centromere and centrosome function and are visible in this image bound to chromosomes within a nucleus of a mitotic cell in prophase (yellow arrow). (**E**) Staining of a melanoma sample using the GE OMX Blaze structured illumination microscope with a 60X/1.42NA objective shows heterogeneity of structural proteins of the nucleus, including as Lamin B and Lamin A/C (indicated by yellow arrows) and part of the nuclear pore complex (NUP98) that measures ~120 nm in total size and indirectly allows the visualization of nuclear pores (indicated by non-continuous staining of NUP98). (**F**) Staining of a patient-derived mouse xenograft breast tumor using the OMX Blaze with a 60x/1.42NA objective shows a spindle in a mitotic cell (beta-tubulin in red) as well as vesicles staining positive for VEGFR2 (in cyan) and punctuate expression of the EGFR in the plasma membrane (in green).

DOI: https://doi.org/10.7554/eLife.31657.006

The following figure supplements are available for figure 2:

*Figure 2 continued on next page*

*Figure 2 continued*

**Figure supplement 1.** Flat-field and shading correction for stitched images.
DOI: https://doi.org/10.7554/eLife.31657.007
**Figure supplement 2.** OMX super-resolution t-CyCIF images.
DOI: https://doi.org/10.7554/eLife.31657.008

collect data with sufficient throughput that large specimens (several square centimeters) can be imaged and analyzed, (v) generate high-resolution data typical of optical microscopy, and (vi) allow investigators to customize the antibody mix to specific questions or tissue types. Among these requirements the last is particularly critical: at the current early stage of development of high dimensional histology, it is essential that individual research groups be able to test the widest possible range of antibodies and antigens in search of those with the greatest scientific and diagnostic value.

This paper describes a method for highly multiplexed fluorescence imaging of tissues, tissue-based cyclic immunofluorescence (t-CyCIF), inspired by a cyclic method first described by *Gerdes et al. (2013)*. t-CyCIF also extends a method we previously described for imaging cells grown in culture (*Lin et al., 2015*). In its current implementation, t-CyCIF assembles up to 60-plex images of FFPE tissue sections via successive rounds of four-channel imaging. t-CyCIF uses widely available reagents, conventional slide scanners and microscopes, manual or automated slide processing and simple protocols. It can, therefore, be implemented in most research or clinical laboratories on existing equipment. Our data suggest that high-dimensional imaging methods using cyclic immunofluorescence have the potential to become a robust and widely-used complement to single-cell genomics, enabling routine analysis of tissue and cancer morphology and phenotypes at single-cell resolution.

## Results

### t-CyCIF enables multiplexed imaging of FFPE tissue and tumor specimens at subcellular resolution

Cyclic immunofluorescence (*Gerdes et al., 2013*) creates highly multiplexed images using an iterative process (a cycle) in which conventional low-plex fluorescence images are repeatedly collected from the same sample and then assembled into a high-dimensional representation. In the implementation described here, samples ~5 μm thick are cut from FFPE blocks, the standard in most histopathology services, followed be dewaxing and antigen retrieval either manually or on automated slide strainers in the usual manner (*Shi et al., 2011*). To reduce auto-fluorescence and non-specific antibody binding, a cycle of 'pre-staining' is performed; this involves incubating the sample with secondary antibodies followed by fluorophore oxidation in a high pH hydrogen peroxide solution in the presence of light ('fluorophore bleaching'). Subsequent t-CyCIF cycles each involve four steps (*Figure 1A*): (i) immuno-staining with antibodies against protein antigens (three antigens per cycle in the implementation described here) (ii) staining with a DNA dye (commonly Hoechst 33342) to mark nuclei and facilitate image registration across cycles (iii) four-channel imaging at low- and high-magnification (iv) fluorophore bleaching followed by a wash step and then another round of immuno-staining. In t-CyCIF, the signal-to-noise ratio often increases with cycle number due to progressive reductions in background intensity over the course of multiple rounds of fluorophore bleaching. This effect is visible in *Figure 1B* as the gradual disappearance of an auto-fluorescent feature (denoted by a dotted white oval and quantified in *Figure 1—figure supplement 1*; see detailed analysis below). When no more t-CyCIF cycles are to be performed, the specimen is stained with H&E to enable conventional histopathology review. Individual image panels are stitched together and registered across cycles followed by image processing and segmentation to identify cells and other structures. t-CyCIF allows for one cycle of indirect immunofluorescence using secondary antibodies. In all other cycles antibodies are directly conjugated to fluorophores, typically Alexa 488, 555 or 647 (for a description of different modes of CyCIF see *Lin et al., 2015*). As an alternative to chemical coupling we have tested the Zenon antibody labeling method (*Tang et al., 2010*) from ThermoFisher in which isotype-specific Fab fragments pre-labeled with fluorophores are bound to primary antibodies to create immune complexes; the immune complexes are then incubated with tissue samples

**Table 1.** Microscopes used in this study and their properties.

| Instrument | Type | Objective | Field of view | Nominal Resolution* |
|---|---|---|---|---|
| RareCyte Cytefinder | Slide Scanner | 10X/0.3 NA | 1.6 × 1.4 mm | 1.06 μm |
| | | 20X/0.8NA | 0.8 × 0.7 mm | 0.40 μm |
| | | 40X/0.6 NA | 0.42 × 0.35 mm | 0.53 μm |
| GE INCell Analyzer 6000 | Confocal | 60X/0.95 NA | 0.22 × 0.22 mm | 0.21 μm |
| GE OMX Blaze | Structured Illumination Microscope | 60 × 1.42 NA | 0.08 × 0.08 mm | 0.11 μm |

*Except in the case of the OMX Blaze, nominal resolution was calculated using the formula (r) = 0.61λ/NA for widefield and (r) = 0.4λ/NA for confocal microscopy with λ = 520 nm. Actual resolution depends on optical properties and thickness of sample, alignment and quality of the optical components in the light path. For structured illumination microscopy, actual resolution depends on accurate matching of immersion oil refractive index with sample in the Cy3 channel and use of an optimal point spread function during reconstruction process. The resolution in other channels will be sub-nominal.

DOI: https://doi.org/10.7554/eLife.31657.009

(*Figure 1—figure supplement 2*). This method is effective with 30–40% of the primary antibodies that we have tested and potentially represents a simple way to label a wide range of primary antibodies with different fluorophores.

Imaging of t-CyCIF samples can be performed on a variety of fluorescent microscopes each of which represent a different tradeoff between data acquisition time, image resolution and sensitivity (*Table 1*). Greater resolution (a higher numerical aperture objective lens) typically corresponds to a smaller field of view and thus, longer acquisition time for large specimens. Imaging of specimens several square centimeters in area at a resolution of ~1 μm is routinely performed on microscopes specialized for scanning slides (slide scanners); we use a CyteFinder system from RareCyte (Seattle WA) configured with 10 × 0.3 NA and 40 × 0.6 NA objectives but have tested scanners from Leica, Nikon and other manufacturers. *Figure 2A–B* show an H&E image of a ~10 × 11 mm metastatic melanoma specimen and a t-CyCIF image assembled from 165 individual image tiles. The assembly process involves stitching sequential image tiles from a single t-CyCIF cycle into one large image panel, flat-fielding to correct for uneven illumination and registration of images from successive t-CyCIF cycles to each other; these procedures were performed using ImageJ, ASHLAR, and BaSiC software as described in materials and methods (*Peng et al., 2017*).

In the t-CyCIF image (*Figure 2B*) tumor cells staining positive for S100 (a melanoma marker in green [*Henze et al., 1997*]) are surrounded by CD45-positive immune cells (CD45RO$^+$ cells in white) and by stromal cells expressing the alpha isoform of smooth muscle actin (α-SMA in red). By zooming in on one tile, single cells can be identified and characterized (*Figure 2C*); in this image, CD4$^+$ and CD8$^+$ T-lymphocytes and proliferating pRB$^+$ positive cells are visible. At 60X resolution on a confocal GE INCell Analyzer 6000, kinetochores stain positive for the phosphorylated form of the Aurora A/B/C kinase and can be counted in a mitotic cell (yellow arrowhead in *Figure 2D*). Nominally super-resolution imaging on a GE OMX Blaze Structured Illumination Microscope (*Carlton et al., 2010*) (using a 60 × 1.42 Plan Apo objective) reveals very fine structural details including differential expression of Lamin isotypes (in a melanoma, *Figure 2E* and *Figure 2—figure supplement 2*) and mitotic spindle fibers (in cells of a xenograft tumor; *Figure 2F* and *Figure 2—figure supplement 2*). These data show that t-CyCIF images have readily interpretable features at the scale of an entire tumor, individual tumor cells and subcellular structures. Little subcellular (or super-resolution) imaging of clinical FFPE specimens has been reported to date (but see *Chen et al., 2015*), but fine subcellular morphology has the potential to provide dramatically greater information than simple integration of antibody intensities across whole cells.

To date, we have tested commercial antibodies against ~200 different proteins for their compatibility with t-CyCIF; these include lineage makers, cytoskeletal proteins, cell cycle regulators, the phosphorylated forms of signaling proteins and kinases, transcription factors, markers of cell state including quiescence, senescence, apoptosis, stress, etc. as well as a variety of non-antibody-based fluorescent stains (*Table 2*). Multiplexing antibodies and stains makes it possible to discriminate among proliferating, quiescent and dying cells, identify tumor and stroma, and collect immuno-phenotypes (*Angelo et al., 2014*; *Giesen et al., 2014*; *Goltsev, 2017*). Use of phospho-specific

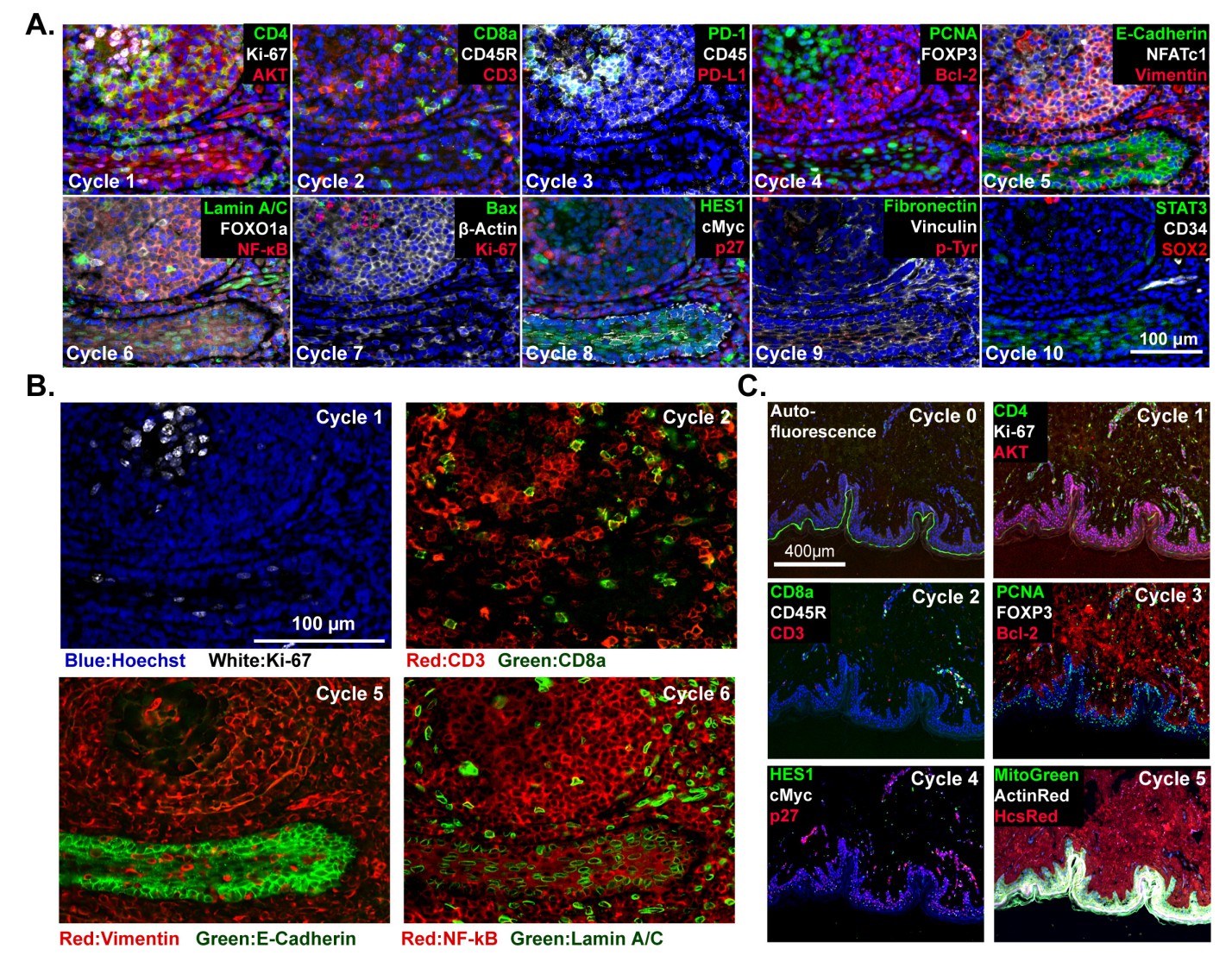

**Figure 3.** t-CyCIF imaging of normal tissues. (**A**) Selected images of a tonsil specimen subjected to 10-cycle t-CyCIF to demonstrate tissue, cellular, and subcellular localization of tissue and immune markers (see **Supplementary file 1** for a list of antibodies). (**B**) Selected cycles from (**A**) demonstrating sub-nuclear features (Ki67 staining, cycle 1), immune cell distribution (cycle 2), structural proteins (E-Cadherin and Vimentin, cycle 5) and nuclear vs. cytosolic localization of transcription factors (NF-kB, cycle 6). (**C**) Five-cycle t-CyCIF of human skin to show the tight localization of some auto-fluorescence signals (Cycle 0), the elimination of these signals after pre-staining (Cycle 1), and the dispersal of rare cell types within a complex layered tissue (see **Supplementary file 1** for a list of the antibodies).

DOI: https://doi.org/10.7554/eLife.31657.010

antibodies and antibodies against proteins that re-localize upon activation (e.g. transcription factors) makes it possible to assay the states of signal transduction networks. For example, in a 10-cycle t-CyCIF analysis of human tonsil (**Figure 3A**) subcellular features such as membrane staining, Ki-67 puncta (Cycle 1), ring-like staining of the nuclear lamina (Cycle 6) and nuclear exclusion of NF-$K$B (Cycle 6) can easily be demonstrated (**Figure 3B**). The five-cycle t-CyCIF data on normal skin in **Figure 3C** shows tight localization of auto-fluorescence (likely melanin) to the epidermis prior to pre-bleaching and images of three non-antibody stains used in the last t-CyCIF cycle: HCS CellMask Red Stain for cytoplasm and nuclei, Actin Red, a Phalloidin-based stain for actin and Mito-tracker Green for mitochondria.

**Table 2.** List of antibodies tested and validated for t-CyCIF.

| Antibody name | Target protein | Performance | Vendor | Catalog no. | Clone | Fluorophore | Research resource Identifier |
|---|---|---|---|---|---|---|---|
| Bax-488 | Bax | * | BioLegend | 633603 | 2D2 | Alexa Fluor 488 | AB_2562171 |
| CD11b-488 | CD11b | * | Abcam | AB204271 | EPR1344 | Alexa Fluor 488 | |
| CD4-488 | CD4 | * | R and D Systems | FAB8165G | Polyclonal | Alexa Fluor 488 | |
| CD8a-488 | CD8 | * | eBioscience | 53-0008-80 | AMC908 | Alexa Fluor 488 | AB_2574412 |
| cJUN-488 | cJUN | * | Abcam | AB193780 | E254 | Alexa Fluor 488 | |
| CK18-488 | Cytokeratin 18 | * | eBioscience | 53-9815-80 | LDK18 | Alexa Fluor 488 | AB_2574480 |
| CK8-FITC | Cytokeratin 8 | * | eBioscience | 11-9938-80 | LP3K | FITC | AB_10548518 |
| CycD1-488 | CycD1 | * | Abcam | AB190194 | EPR2241 | Alexa Fluor 488 | |
| Ecad-488 | E-Cadherin | * | CST | 3199 | 24E10 | Alexa Fluor 488 | AB_10691457 |
| EGFR-488 | EGFR | * | CST | 5616 | D38B1 | Alexa Fluor 488 | AB_10691853 |
| EpCAM-488 | EpCAM | * | CST | 5198 | VU1D9 | Alexa Fluor 488 | AB_10692105 |
| HES1-488 | HES1 | * | Abcam | AB196328 | EPR4226 | Alexa Fluor 488 | |
| Ki67-488 | Ki67 | * | CST | 11882 | D3B5 | Alexa Fluor 488 | AB_2687824 |
| LaminA/C-488 | Lamin A/C | * | CST | 8617 | 4C11 | Alexa Fluor 488 | AB_10997529 |
| LaminB1-488 | Lamin B1 | * | Abcam | AB194106 | EPR8985(B) | Alexa Fluor 488 | |
| mCD3E-FITC | ms_CD3E | * | BioLegend | 100306 | 145–2 C11 | FITC | AB_312671 |
| mCD4-488 | ms_CD4 | * | BioLegend | 100532 | RM4-5 | Alexa Fluor 488 | AB_493373 |
| MET-488 | c-MET | * | CST | 8494 | D1C2 | Alexa Fluor 488 | AB_10999405 |
| mF4/80-488 | ms_F4/80 | * | BioLegend | 123120 | BM8 | Alexa Fluor 488 | AB_893479 |
| MITF-488 | MITF | * | Abcam | AB201675 | D5 | Alexa Fluor 488 | |
| Ncad-488 | N-Cadherin | * | BioLegend | 350809 | 8C11 | Alexa Fluor 488 | AB_11218797 |
| p53-488 | p53 | * | CST | 5429 | 7F5 | Alexa Fluor 488 | AB_10695458 |
| PCNA-488 | PCNA | * | CST | 8580 | PC10 | Alexa Fluor 488 | AB_11178664 |
| PD1-488 | PD1 | * | CST | 15131 | D3W4U | Alexa Fluor 488 | |
| PDI-488 | PDI | * | CST | 5051 | C81H6 | Alexa Fluor 488 | AB_10950503 |
| pERK-488 | pERK(T202/Y204) | * | CST | 4344 | D13.14.4E | Alexa Fluor 488 | AB_10695876 |
| pNDG1-488 | pNDG1(T346) | * | CST | 6992 | D98G11 | Alexa Fluor 488 | AB_10827648 |

*Table 2 continued on next page*

Table 2 continued

| Antibody name | Target protein | Performance | Vendor | Catalog no. | Clone | Fluorophore | Research resource Identifier |
|---|---|---|---|---|---|---|---|
| POL2A-488 | POL2A | * | Novus Biologicals | NB200-598AF488 | 4H8 | Alexa Fluor 488 | AB_2167465 |
| pS6(S240/244)−488 | pS6(240/244) | * | CST | 5018 | D68F8 | Alexa Fluor 488 | AB_10695861 |
| S100a-488 | S100alpha | * | Abcam | AB207367 | EPR5251 | Alexa Fluor 488 | |
| SQSTM1-488 | SQSTM1/p62 | * | CST | 8833 | D1D9E3 | Alexa Fluor 488 | |
| STAT3-488 | STAT3 | * | CST | 14047 | B3Z2G | Alexa Fluor 488 | |
| Survivin-488 | Survivin | * | CST | 2810 | 71G4B7 | Alexa Fluor 488 | AB_10691462 |
| Catenin-488 | β-Catenin | * | CST | 2849 | L54E2 | Alexa Fluor 488 | AB_10693296 |
| Actin-555 | Actin | * | CST | 8046 | 13E5 | Alexa Fluor 555 | AB_11179208 |
| CD11c-570 | CD11c | * | eBioscience | 41-9761-80 | 118/A5 | eFluor 570 | AB_2573632 |
| CD3D-555 | CD3D | * | Abcam | AB208514 | EP4426 | Alexa Fluor 555 | |
| CD4-570 | CD4 | * | eBioscience | 41-2444-80 | N1UG0 | eFluor 570 | AB_2573601 |
| CD45-PE | CD45 | * | R and D Systems | FAB1430P-100 | 2D1 | PE | AB_2237898 |
| CK7-555 | Cytokeratin 7 | * | Abcam | AB209601 | EPR17078 | Alexa Fluor 555 | |
| cMYC-555 | cMYC | * | Abcam | AB201780 | Y69 | Alexa Fluor 555 | |
| E2F1-555 | E2F1 | * | Abcam | AB208078 | EPR3818(3) | Alexa Fluor 555 | |
| Ecad-555 | E-Cadherin | * | CST | 4295 | 24E10 | Alexa Fluor 555 | |
| EpCAM-PE | EpCAM | * | BioLegend | 324205 | 9C4 | PE | AB_756079 |
| FOXO1a-555 | FOXO1a | * | Abcam | AB207244 | EP927Y | Alexa Fluor 555 | |
| FOXP3-570 | FOXP3 | * | eBioscience | 41-4777-80 | 236A/E7 | eFluor 570 | AB_2573608 |
| GFAP-570 | GFAP | * | eBioscience | 41-9892-80 | GA5 | eFluor 570 | AB_2573655 |
| HSP90-PE | HSP90b | * | Abcam | AB115641 | Polyclonal | PE | AB_10936222 |
| KAP1-594 | KAP1 | * | BioLegend | 619304 | 20A1 | Alexa Fluor 594 | AB_2563298 |
| Keratin-555 | pan-Keratin | * | CST | 3478 | C11 | Alexa Fluor 555 | AB_10829040 |
| Keratin-570 | pan-Keratin | * | eBioscience | 41-9003-80 | AE1/AE3 | eFluor 570 | AB_11217482 |
| Ki67-570 | Ki67 | * | eBioscience | 41-5699-80 | 20Raj1 | eFluor 570 | AB_11220088 |
| LC3-555 | LC3 | * | CST | 13173 | D3U4C | Alexa Fluor 555 | |
| MAP2-570 | MAP2 | * | eBioscience | 41-9763-80 | AP20 | eFluor 570 | AB_2573634 |
| pAUR-555 | pAUR1/2/3(T288/T2 | * | CST | 13464 | D13A11 | Alexa Fluor 555 | |
| pCHK2-PE | pChk2(T68) | * | CST | 12812 | C13C1 | PE | |
| PDL1-555 | PD-L1/CD274 | * | Abcam | AB213358 | 28–8 | Alexa Fluor 555 | |

Table 2 continued on next page

Table 2 continued

| Antibody name | Target protein | Performance | Vendor | Catalog no. | Clone | Fluorophore | Research resource Identifier |
|---|---|---|---|---|---|---|---|
| pH3-555 | pH3(S10) | * | CST | 3475 | D2C8 | Alexa Fluor 555 | AB_10694639 |
| pRB-555 | pRB(S807/811) | * | CST | 8957 | D20B12 | Alexa Fluor 555 | |
| pS6(235/236)–555 | pS6(235/236) | * | CST | 3985 | D57.2.2E | Alexa Fluor 555 | AB_10693792 |
| pSRC-PE | pSRC(Y418) | * | eBioscience | 12-9034-41 | SC1T2M3 | PE | AB_2572680 |
| S6-555 | S6 | * | CST | 6989 | 54D2 | Alexa Fluor 555 | AB_10828226 |
| SQSTM1-555 | SQSTM1/p62 | * | Abcam | AB203430 | EPR4844 | Alexa Fluor 555 | |
| VEGFR2-555 | VEGFR2 | * | CST | 12872 | D5B1 | Alexa Fluor 555 | |
| VEGFR2-PE | VEGFR2 | * | CST | 12634 | D5B1 | PE | |
| Vimentin-555 | Vimentin | * | CST | 9855 | D21H3 | Alexa Fluor 555 | AB_10859896 |
| Vinculin-570 | Vinculin | * | eBioscience | 41-9777-80 | 7F9 | eFluor 570 | AB_2573646 |
| gH2ax-PE | gH2ax | * | BioLegend | 613412 | 2F3 | PE | AB_2616871 |
| AKT-647 | AKT | * | CST | 5186 | C67E7 | Alexa Fluor 647 | AB_10695877 |
| aSMA-660 | aSMA | * | eBioscience | 50-9760-80 | 1A4 | eFluor 660 | AB_2574361 |
| B220-647 | CD45R/B220 | * | BioLegend | 103226 | RA3-6B2 | Alexa Fluor 647 | AB_389330 |
| Bcl2-647 | Bcl2 | * | BioLegend | 658705 | 100 | Alexa Fluor 647 | AB_2563279 |
| Catenin-647 | Beta-Catenin | * | CST | 4627 | L54E2 | Alexa Fluor 647 | AB_10691326 |
| CD20-660 | CD20 | * | eBioscience | 50-0202-80 | L26 | eFluor 660 | AB_11151691 |
| CD45-647 | CD45 | * | BioLegend | 304020 | HI30 | Alexa Fluor 647 | AB_493034 |
| CD8a-660 | CD8 | * | eBioscience | 50-0008-80 | AMC908 | eFluor 660 | AB_2574148 |
| CK5-647 | Cytokeratin 5 | * | Abcam | AB193895 | EP1601Y | Alexa Fluor 647 | |
| ColIV-647 | Collagen IV | * | eBioscience | 51-9871-80 | 1042 | Alexa Fluor 647 | AB_10854267 |
| COXIV-647 | COXIV | * | CST | 7561 | 3E11 | Alexa Fluor 647 | AB_10994876 |
| cPARP-647 | cPARP | * | CST | 6987 | D64E10 | Alexa Fluor 647 | AB_10858215 |
| FOXA2-660 | FOXA2 | * | eBioscience | 50-4778-82 | 3C10 | eFluor 660 | AB_2574221 |
| FOXP3-647 | FOXP3 | * | BioLegend | 320113 | 206D | Alexa Fluor 647 | AB_439753 |
| gH2ax-647 | H2ax(S139) | * | CST | 9720 | 20E3 | Alexa Fluor 647 | AB_10692910 |
| gH2ax-647 | H2ax(S139) | * | BioLegend | 613407 | 2F3 | Alexa Fluor 647 | AB_2114994 |
| HES1-647 | HES1 | * | Abcam | AB196577 | EPR4226 | Alexa Fluor 647 | |
| Ki67-647 | Ki67 | * | CST | 12075 | D3B5 | Alexa Fluor 647 | |

Table 2 continued on next page

*Table 2 continued*

| Antibody name | Target protein | Performance | Vendor | Catalog no. | Clone | Fluorophore | Research resource Identifier |
|---|---|---|---|---|---|---|---|
| Ki67-647 | Ki67 | * | BioLegend | 350509 | Ki-67 | Alexa Fluor 647 | AB_10900810 |
| mCD45-647 | ms_CD45 | * | BioLegend | 103124 | 30-F11 | Alexa Fluor 647 | AB_493533 |
| mCD4-647 | ms_CD4 | * | BioLegend | 100426 | GK1.5 | Alexa Fluor 647 | AB_493519 |
| mEPCAM-647 | ms_EPCAM | * | BioLegend | 118211 | G8.8 | Alexa Fluor 647 | AB_1134104 |
| MHCI-647 | MHCI/HLAA | * | Abcam | AB199837 | EP1395Y | Alexa Fluor 647 | |
| MHCII-647 | MHCII | * | Abcam | AB201347 | EPR11226 | Alexa Fluor 647 | |
| mLy6C-647 | ms_Ly6C | * | BioLegend | 128009 | HK1.4 | Alexa Fluor 647 | AB_1236551 |
| mTOR-647 | mTOR | * | CST | 5048 | 7C10 | Alexa Fluor 647 | AB_10828101 |
| NFkB-647 | NFkB (p65) | * | Abcam | AB190589 | E379 | Alexa Fluor 647 | |
| NGFR-647 | NGFR/CD271 | * | Abcam | AB195180 | EP1039Y | Alexa Fluor 647 | |
| NUP98-647 | NUP98 | * | CST | 13393 | C39A3 | Alexa Fluor 647 | |
| p21-647 | p21 | * | CST | 8587 | 12D1 | Alexa Fluor 647 | AB_10892861 |
| p27-647 | p27 | * | Abcam | AB194234 | Y236 | Alexa Fluor 647 | |
| pATM-660 | pATM(S1981) | * | eBioscience | 50-9046-41 | 10H11.E12 | eFluor 660 | AB_2574312 |
| PAX8-647 | PAX8 | * | Abcam | AB215953 | EPR18715 | Alexa Fluor 647 | |
| PDL1-647 | PD-L1/CD274 | * | CST | 15005 | E1L3N | Alexa Fluor 647 | |
| pMK2-647 | pMK2(T334) | * | CST | 4320 | 27B7 | Alexa Fluor 647 | AB_10695401 |
| pmTOR-660 | pmTOR(S2448) | * | eBioscience | 50-9718-41 | MRRBY | eFluor 660 | AB_2574351 |
| pS6_235–647 | pS6(S235/S236) | * | CST | 4851 | D57.2.2E | Alexa Fluor 647 | AB_10695457 |
| pSTAT3-647 | pSTAT3(Y705) | * | CST | 4324 | D3A7 | Alexa Fluor 647 | AB_10694637 |
| pTyr-647 | p-Tyrosine | * | CST | 9415 | p-Tyr-100 | Alexa Fluor 647 | AB_10693160 |
| S100A4-647 | S100A4 | * | Abcam | AB196168 | EPR2761(2) | Alexa Fluor 647 | |
| Survivin-647 | Survivin | * | CST | 2866 | 71G4B7 | Alexa Fluor 647 | AB_10698609 |
| TUBB3-647 | TUBB3 | * | BioLegend | 657405 | AA10 | Alexa Fluor 647 | AB_2563609 |
| Tubulin-647 | beta-Tubulin | * | CST | 3624 | 9F3 | Alexa Fluor 647 | AB_10694204 |
| Vimentin-647 | Vimentin | * | BioLegend | 677807 | O91D3 | Alexa Fluor 647 | AB_2616801 |
| anti-14-3-3 | 14-3-3 | * | Santa Cruz | SC-629-G | Polyclonal | N/D | AB_630820 |
| anti-53BP1 | 53BP1 | * | Bethyl | A303-906A | Polyclonal | N/D | AB_2620256 |

*Table 2 continued on next page*

*Table 2 continued*

| Antibody name | Target protein | Performance | Vendor | Catalog no. | Clone | Fluorophore | Research resource Identifier |
|---|---|---|---|---|---|---|---|
| anti-5HMC | 5HMC | * | Active Motif | 39769 | Polyclonal | N/D | AB_10013602 |
| anti-CD11b | CD11b | * | Abcam | AB133357 | EPR1344 | N/D | AB_2650514 |
| anti-CD2 | CD2 | * | Abcam | AB37212 | Polyclonal | N/D | AB_726228 |
| anti-CD20 | CD20 | * | Dako | M0755 | L26 | N/D | AB_2282030 |
| anti-CD3 | CD3 | * | Dako | A0452 | Polyclonal | N/D | AB_2335677 |
| anti-CD4 | CD4 | * | Dako | M7310 | 4B12 | N/D | |
| anti-CD45RO | CD45RO | * | Dako | M0742 | UCHL1 | N/D | AB_2237910 |
| anti-CD8 | CD8 | * | Dako | M7103 | C8/144B | N/D | AB_2075537 |
| anti-CycA2 | CycA2 | * | Abcam | AB38 | E23.1 | N/D | AB_304084 |
| anti-ET1 | ET-1 | * | Abcam | AB2786 | TR.ET.48.5 | N/D | AB_303299 |
| anti-FAP | FAP | * | eBioscience | BMS168 | F11-24 | N/D | AB_10597443 |
| anti-FOXP3 | FOXP3 | * | BioLegend | 320102 | 206D | N/D | AB_430881 |
| anti-LAMP2 | LAMP2 | * | Abcam | AB25631 | H4B4 | N/D | AB_470709 |
| anti-MCM6 | MCM6 | * | Santa Cruz | SC-9843 | Polyclonal | N/D | AB_2142543 |
| anti-PAX8 | PAX8 | * | Abcam | AB191870 | EPR18715 | N/D | |
| anti-PD1 | PD1 | * | CST | 86163 | D4W2J | N/D | |
| anti-pEGFR | pEGFR(Y1068) | * | CST | 3777 | D7A5 | N/D | AB_2096270 |
| anti-pERK | pERK(T202/Y204) | * | CST | 4370 | D13.14.4E | N/D | AB_2315112 |
| anti-pRB | pRB(S807/811) | * | Santa Cruz | SC-16670 | Polyclonal | N/D | AB_655250 |
| anti-pRPA32 | pRPA32 (S4/S8) | * | Bethyl | IHC-00422 | Polyclonal | N/D | AB_1659840 |
| anti-pSTAT3 | pSTAT3 | ** | CST | 9145 | D3A7 | N/D | AB_2491009 |
| anti-pTyr | pTyr | * | CST | 9411 | p-Tyr-100 | N/D | AB_331228 |
| anti-RPA32 | RPA32 | * | Bethyl | IHC-00417 | Polyclonal | N/D | AB_1659838 |
| anti-TPCN2 | TPCN2 | * | NOVUSBIO | NBP1-86923 | Polyclonal | N/D | AB_11021735 |
| anti-VEGFR1 | VEGFR1/FLT1 | * | Santa Cruz | SC-31173 | Polyclonal | N/D | AB_2106885 |
| Abeta-488 | Beta-Amyloid (1-16) | † | BioLegend | 803013 | 6E10 | Alexa Fluor 488 | AB_2564765 |
| BRAF-FITC | B-RAF | † | Abcam | ab175637 | K21-F | FITC | |
| BrdU-488 | BrdU | † | BioLegend | 364105 | 3D4 | Alexa Fluor 488 | AB_2564499 |
| cCasp3-488 | cCasp3 | † | R and D Systems | IC835G-025 | 269518 | Alexa Fluor 488 | |
| CD11b-488 | CD11b | † | BioLegend | 101219 | M1/70 | Alexa Fluor 488 | AB_493545 |
| CD123-488 | CD123 | † | BioLegend | 306035 | 6H6 | Alexa Fluor 488 | AB_2629569 |
| CD49b-FITC | CD49b | † | BioLegend | 359305 | P1E6-C5 | FITC | AB_2562530 |
| CD69-FITC | CD69 | † | BioLegend | 310904 | FN50 | FITC | AB_314839 |
| CD71-FITC | CD71 | † | BioLegend | 334103 | CY1G4 | FITC | AB_1236432 |
| CD80-FITC | CD80 | † | R and D Systems | FAB140F | 37711 | FITC | AB_357027 |
| CD8a-488 | CD8a | † | eBioscience | 53-0086-41 | OKT8 | Alexa Fluor 488 | AB_10547060 |
| CDC2-FITC | CDC2/p34 | † | Santa Cruz | SC-54 FITC | 17 | FITC | AB_627224 |
| CycB1-FITC | CycB1 | † | Santa Cruz | SC-752 FITC | Polyclonal | FITC | AB_2072134 |

*Table 2 continued on next page*

*Table 2 continued*

| Antibody name | Target protein | Performance | Vendor | Catalog no. | Clone | Fluorophore | Research resource Identifier |
|---|---|---|---|---|---|---|---|
| FN-488 | Fibronection | † | Abcam | AB198933 | F1 | Alexa Fluor 488 | |
| IFNG-488 | Interferon-Gamma | † | BioLegend | 502517 | 4S.B3 | Alexa Fluor 488 | AB_493030 |
| IL1-FITC | IL1 | † | BioLegend | 511705 | H1b-98 | FITC | AB_1236434 |
| IL6-FITC | IL6 | † | BioLegend | 501103 | MQ2-13A5 | FITC | AB_315151 |
| mCD31-FITC | ms_CD31 | † | eBioscience | 11-0311-82 | 390 | FITC | AB_465012 |
| mCD8a-488 | ms_CD8a | † | BioLegend | 100726 | 53–6.7 | Alexa Fluor 488 | AB_493423 |
| Nestin-488 | Nestin | † | eBioscience | 53-9843-80 | 10C2 | Alexa Fluor 488 | AB_1834347 |
| NeuN-488 | NeuN | † | Millipore | MAB377X | A60 | Alexa Fluor 488 | AB_2149209 |
| PR-488 | PR/PGR | † | Abcam | AB199224 | YR85 | Alexa Fluor 488 | |
| Snail1-488 | Snail1 | † | eBioscience | 53-9859-80 | 20C8 | Alexa Fluor 488 | AB_2574482 |
| TGFB-FITC | TGFB1 | † | BioLegend | 349605 | TW4-2F8 | FITC | AB_10679043 |
| TNFa-488 | TNFa | † | BioLegend | 502917 | MAb11 | Alexa Fluor 488 | AB_493122 |
| AR-555 | AR | † | CST | 8956 | D6F11 | Alexa Fluor 555 | AB_11129223 |
| CD11a-PE | CD11a | † | BioLegend | 301207 | HI111 | PE | AB_314145 |
| CD11b-555 | CD11b | † | Abcam | AB206616 | EPR1344 | Alexa Fluor 555 | |
| CD131-PE | CD131 | † | BD | 559920 | JORO50 | PE | AB_397374 |
| CD14-PE | CD14 | † | eBioscience | 12–0149 | 61D3 | PE | AB_10597598 |
| CD1a-PE | CD1a | † | BioLegend | 300105 | HI149 | PE | AB_314019 |
| CD1c-PE | CD1c | † | BioLegend | 331505 | L161 | PE | AB_1089000 |
| CD20-PE | CD20 | † | BioLegend | 302305 | 2H7 | PE | AB_314253 |
| CD23-PE | CD23 | † | eBioscience | 12-0232-81 | B3B4 | PE | AB_465592 |
| CD31-PE | CD31 | † | eBioscience | 12-0319-41 | WM-59 | PE | AB_10670623 |
| CD31-PE | CD31 | † | R and D Systems | FAB3567P-025 | 9G11 | PE | AB_2279388 |
| CD34-PE | CD34 | † | Abcam | AB30377 | QBEND/10 | PE | AB_726407 |
| CD45R-e570 | CD45R/B220 | † | eBioscience | 41-0452-80 | RA3-6B2 | eFluor 570 | AB_2573598 |
| CD71-PE | CD71 | † | eBioscience | 12-0711-81 | R17217 | PE | AB_465739 |
| CD86-PE | CD86 | † | BioLegend | 305405 | IT2.2 | PE | AB_314525 |
| CK19-570 | Cytokeratin 19 | † | eBioscience | 41-9898-80 | BA17 | eFluor 570 | AB_11218678 |
| HER2-570 | HER2 | † | eBioscience | 41-9757-80 | MJD2 | eFluor 570 | AB_2573628 |
| IL3-PE | IL3 | † | BD | 554383 | MP2-8F8 | PE | AB_395358 |
| NFATc1-PE | NFATc1 | † | BioLegend | 649605 | 7A6 | PE | AB_2562546 |
| PDL1-PE | PD-L1/CD274 | † | BioLegend | 329705 | 29E.2A3 | PE | AB_940366 |
| pMAPK (T202/Y204) | pERK1/2(T202/Y20 | † | CST | 14095 | 197G2 | PE | |
| pMAPK (Y204/Y187) | pERK1/2(Y204/Y18 | † | CST | 75165 | D1H6G | PE | |
| pSTAT1-PE | pSTAT1(Y705) | † | BioLegend | 686403 | A15158B | PE | AB_2616938 |

*Table 2 continued on next page*

*Table 2 continued*

| Antibody name | Target protein | Performance | Vendor | Catalog no. | Clone | Fluorophore | Research resource Identifier |
|---|---|---|---|---|---|---|---|
| ABCC1-647 | ABCC1 | † | BioLegend | 370203 | QCRL-2 | Alexa Fluor 647 | AB_2566664 |
| AnnexinV-674 | N/D | † | BioLegend | 640911 | NA | Alexa Fluor 647 | AB_2561293 |
| CD103-647 | CD103 | † | BioLegend | 350209 | Ber-ACT8 | Alexa Fluor 647 | AB_10640870 |
| CD25-647 | CD25 | † | BioLegend | 302617 | BC96 | Alexa Fluor 647 | AB_493046 |
| CD31-APC | CD31 | † | eBioscience | 17-0319-41 | WM-59 | APC | AB_10853188 |
| CD68-APC | CD68 | † | BioLegend | 333809 | Y1/82A | APC | AB_10567107 |
| CD8a-647 | CD8a | † | BioLegend | 344725 | SK1 | Alexa Fluor 647 | AB_2563451 |
| CD8a-647 | CD8a | † | R and D Systems | FAB1509R-025 | 37006 | Alexa Fluor 647 | |
| CycE-660 | CycE | † | eBioscience | 50-9714-80 | HE12 | eFluor 660 | AB_2574350 |
| HIF1-647 | HIF1 | † | BioLegend | 359705 | 546–16 | Alexa Fluor 647 | AB_2563331 |
| HP1-647 | HP1 | † | Abcam | AB198391 | EPR5777 | Alexa Fluor 647 | |
| mCD123-APC | ms_CD123 | † | eBioscience | 17-1231-81 | 5B11 | APC | AB_891363 |
| NGFR-647 | NGFR/CD271 | † | BD | 560326 | C40-1457 | Alexa Fluor 647 | AB_1645403 |
| pBTK-660 | pBTK(Y551/Y511) | † | eBioscience | 50-9015-80 | M4G3LN | eFluor 660 | AB_2574306 |
| PD1-647 | PD1 | † | Abcam | AB201825 | EPR4877 (2) | Alexa Fluor 647 | |
| PR-660 | PR/PGR | † | eBioscience | 50-9764-80 | KMC912 | eFluor 660 | AB_2574363 |
| RUNX3-660 | RUNX3 | † | eBioscience | 50-9817-80 | R3-5G4 | eFluor 660 | AB_2574383 |
| SOX2-647 | SOX2 | † | Abcam | AB192075 | Polyclonal | Alexa Fluor 647 | |
| anti-53BP1 | 53BP1 | † | Millipore | MAB3802 | BP13 | N/D | AB_2206767 |
| anti-Axl | Axl | † | R and D | AF154 | Polyclonal | N/D | AB_354852 |
| anti-CD11b | CD11b | † | Abcam | AB52478 | EP1345Y | N/D | AB_868788 |
| anti-CD8a | CD8 | † | eBioscience | 14-0085-80 | C8/144B | N/D | AB_11151339 |
| anti-CEP170 | CEP170 | † | Abcam | AB72505 | Polyclonal | N/D | AB_1268101 |
| anti-cMYC | cMYC | † | BioLegend | 626801 | 9E10 | N/D | AB_2235686 |
| anti-CPS1 | CPS1 | † | Abcam | AB129076 | EPR7493-3 | N/D | AB_11156290 |
| anti-E2F1 | E2F1 | † | ThermoFisher | MS-879-P1 | KH95 | N/D | AB_143934 |
| anti-eEF2K | eEF2K | † | Santa Cruz | SC-21642 | K-19 | N/D | AB_640043 |
| anti-Emil1 | Emil1 | † | Abcam | AB212397 | EMIL/1176 | N/D | |
| anti-FKHRL1 | FKHRL1 | † | Santa Cruz | SC-9812 | Polyclonal | N/D | AB_640608 |
| anti-FLAG | FLAG | † | Sigma | F1804 | M2 | N/D | AB_262044 |
| anti-GranB | Granzyme_B | † | Dako | M7235 | M7235 | N/D | AB_2114697 |
| anti-HMB45 | HMB45 | † | Abcam | AB732 | HMB45 + M2-7C10 + M2-9E3 | N/D | AB_305844 |
| anti-HSP90b | HSP90b | † | Santa Cruz | SC-1057 | D-19 | N/D | AB_2121392 |
| anti-IL2Ra | IL2Ra | † | Abcam | AB128955 | EPR6452 | N/D | AB_11141054 |
| anti-LAMP2 | LAMP2 | † | R and D | AF6228 | Polyclonal | N/D | AB_10971818 |

*Table 2 continued on next page*

*Table 2 continued*

| Antibody name | Target protein | Performance | Vendor | Catalog no. | Clone | Fluorophore | Research resource Identifier |
|---|---|---|---|---|---|---|---|
| anti-MITF | MITF | † | Abcam | AB12039 | C5 | N/D | AB_298801 |
| anti-Ncad | N-Cadherin | † | Abcam | AB18203 | Polyclonal | N/D | AB_444317 |
| anti-NCAM | NCAM | † | Abcam | AB6123 | ERIC-1 | N/D | AB_2149537 |
| anti-NF1 | NF1 | † | Abcam | AB178323 | McNFn27b | N/D | |
| anti-pCTD | Pol II CTD(S2) | † | Active Motif | 61083 | 3E10 | N/D | AB_2687450 |
| anti-PD1 | PD1 | † | CST | 43248 | EH33 | N/D | |
| anti-pTuberin | pTuberin(S664) | † | Abcam | AB133465 | EPR8202 | N/D | AB_11157389 |
| anti-S100 | S100 | † | Dako | Z0311 | Polyclonal | N/D | AB_10013383 |
| anti-SIRT3 | SIRT3 | † | CST | 2627 | C73E3 | N/D | AB_2188622 |
| anti-TIA1 | TIA1 | † | Santa Cruz | SC-1751 | Polyclonal | N/D | AB_2201433 |
| anti-TLR3 | TLR3 | † | Santa Cruz | SC-8691 | Polyclonal | N/D | AB_2240700 |
| anti-TNFa | TNFa | † | Abcam | AB11564 | MP6-XT3 | N/D | AB_298170 |
| anti-TPCN2 | TPCN2 | † | Abcam | AB119915 | Polyclonal | N/D | AB_10903692 |
| CD11a-FITC | CD11a | ‡ | eBioscience | 11-0119-41 | HI111 | FITC | AB_10597888 |
| CD20-FITC | CD20 | ‡ | BioLegend | 302303 | 2H7 | FITC | AB_314251 |
| CD2-FITC | CD2 | ‡ | BioLegend | 300206 | RPA-2.10 | FITC | AB_314030 |
| CD45RO-488 | CD45RO | ‡ | BioLegend | 304212 | UCHL1 | Alexa Fluor 488 | AB_528823 |
| CD8a-488 | CD8 | ‡ | BioLegend | 301024 | RPA-T8 | Alexa Fluor 488 | AB_2561282 |
| cJUN-FITC | cJUN | ‡ | Santa Cruz | SC-1694 FITC | Polyclonal | FITC | AB_631263 |
| CXCR5-FITC | CXCR5 | ‡ | BioLegend | 356913 | J252D4 | FITC | AB_2561895 |
| Ecad-FITC | Ecad | ‡ | BioLegend | 324103 | 67A4 | FITC | AB_756065 |
| FOXP3-488 | FOXP3 | ‡ | BioLegend | 320011 | 150D | Alexa Fluor 488 | AB_439747 |
| MITF-488 | MITF | ‡ | Novus Biologicals | NB100-56561AF488 | 21D1418 | Alexa Fluor 488 | AB_838580 |
| NCAM-488 | NCAM/CD56 | ‡ | Abcam | AB200333 | EPR2566 | Alexa Fluor 488 | |
| NCAM-FITC | NCAM/CD56 | ‡ | ThermoFisher | 11-0566-41 | TULY56 | FITC | AB_2572458 |
| NGFR-FITC | NGFR/CD271 | ‡ | BioLegend | 345103 | ME20.4 | FITC | AB_1937226 |
| PD1-488 | PD-1 | ‡ | BioLegend | 367407 | NAT105 | Alexa Fluor 488 | AB_2566677 |
| PD1-488 | PD-1 | ‡ | BioLegend | 329935 | EH12.2H7 | Alexa Fluor 488 | AB_2563593 |
| pERK-488 | pERK(T202/Y204) | ‡ | CST | 4374 | E10 | Alexa Fluor 488 | AB_10705598 |
| pERK-488 | pERK(T202/Y204) | ‡ | CST | 4780 | 137F5 | Alexa Fluor 488 | AB_10705598 |
| S100A4-FITC | S100A4 | ‡ | BioLegend | 370007 | NJ-4F3-D1 | FITC | AB_2572073 |
| SOX2-488 | SOX2 | ‡ | BioLegend | 656109 | 14A6A34 | Alexa Fluor 488 | AB_2563956 |
| CD133-PE | CD133 | ‡ | eBioscience | 12-1338-41 | TMP4 | PE | AB_1582258 |
| cMyc-TRITC | cMYC | ‡ | Santa Cruz | SC-40 TRITC | 9E10 | TRITC | AB_627268 |
| cPARP-555 | cPARP | ‡ | CST | 6894 | D64E10 | Alexa Fluor 555 | AB_10830735 |
| CTLA4-PE | CTLA4 | ‡ | BioLegend | 369603 | BNI3 | PE | AB_2566796 |

*Table 2 continued on next page*

*Table 2 continued*

| Antibody name | Target protein | Performance | Vendor | Catalog no. | Clone | Fluorophore | Research resource Identifier |
|---|---|---|---|---|---|---|---|
| GATA3-594 | GATA3 | ‡ | BioLegend | 653816 | 16E10A23 | Alexa Fluor 594 | AB_2563353 |
| GFAP-Cy3 | GFAP | ‡ | Millipore | MAB3402C3 | NA | Cy3 | AB_11213580 |
| Oct4-555 | OCT_4 | ‡ | CST | 4439 | C30A3 | Alexa Fluor 555 | AB_10922586 |
| p21-555 | p21 | ‡ | CST | 8493 | 12D1 | Alexa Fluor 555 | AB_10860074 |
| PD1-PE | PD1 | ‡ | BioLegend | 329905 | EH12.2H7 | PE | AB_940481 |
| PDGFRb-555 | PDGFRb | ‡ | Abcam | AB206874 | Y92 | Alexa Fluor 555 | |
| pSTAT1-555 | pSTAT1 | ‡ | CST | 8183 | 58D6 | Alexa Fluor 555 | AB_10860600 |
| TIM1-PE | TIM1 | ‡ | BioLegend | 353903 | 1D12 | PE | AB_11125165 |
| cCasp3-647 | cCasp3 | ‡ | CST | 9602 | D3E9 | Alexa Fluor 647 | AB_2687881 |
| CD103-APC | CD103 | ‡ | eBioscience | 17-1038-41 | B-Ly7 | APC | AB_10669816 |
| CD3-647 | CD3 | ‡ | BioLegend | 300422 | UCHT1 | Alexa Fluor 647 | AB_493092 |
| CD3-660 | CD3 | ‡ | eBioscience | 50-0037-41 | OKT3 | eFluor 660 | AB_2574150 |
| CD3-APC | CD3 | ‡ | eBioscience | 17-0038-41 | UCHT1 | APC | AB_10804761 |
| CD45RO-APC | CD45RO | ‡ | BioLegend | 304210 | UCHL1 | APC | AB_314426 |
| ER-647 | ER | ‡ | Abcam | AB205851 | EPR4097 | Alexa Fluor 647 | |
| FOXO3a-647 | FOXO3a | ‡ | Abcam | AB196539 | EP1949Y | Alexa Fluor 647 | |
| GZMA-e660 | Granzyme A | ‡ | ThermoFisher | 50-9177-41 | CB9 | eFluor 660 | AB_2574330 |
| GZMB-647 | Granzyme_B | ‡ | BioLegend | 515405 | GB11 | Alexa Fluor 647 | AB_2294995 |
| GZMB-APC | Granzyme_B | ‡ | R and D Systems | IC29051A | 356412 | APC | AB_894691 |
| HER2-647 | HER2 | ‡ | BioLegend | 324412 | 24D2 | Alexa Fluor 647 | AB_2262300 |
| mCD49b-647 | ms_CD49b | ‡ | BioLegend | 103511 | HMα2 | Alexa Fluor 647 | AB_528830 |
| NCAM-647 | NCAM/CD56 | ‡ | BioLegend | 362513 | 5.1H11 | Alexa Fluor 647 | AB_2564086 |
| NCAM-e660 | NCAM/CD56 | ‡ | ThermoFisher | 50-0565-80 | 5tukon56 | eFluor 660 | AB_2574160 |
| pAKT-647 | pAKT | ‡ | CST | 4075 | D9E | Alexa Fluor 647 | AB_10691856 |
| pERK-647 | pERK (T202/Y204) | ‡ | CST | 4375 | E10 | Alexa Fluor 647 | AB_10706777 |
| pERK-647 | pERK (T202/Y204) | ‡ | BioLegend | 369503 | 6B8B69 | Alexa Fluor 647 | AB_2571895 |
| pIKBa-660 | pIKBa | ‡ | eBioscience | 50-9035-41 | RILYB3R | eFluor 660 | AB_2574310 |
| YAP-647 | YAP | ‡ | CST | 38707S | D8H1X | Alexa Fluor 647 | |
| anit-FANCD2 | FANCD2 | ‡ | Bethyl | IHC-00624 | Polyclonal | N/D | AB_10752755 |
| anit-pcJUN | p-cJUN | ‡ | Santa Cruz | SC-822 | KM-1 | N/D | AB_627262 |
| anti-AXL | AXL | ‡ | CST | 8661 | C89E7 | N/D | AB_11217435 |
| anti-CXCR5 | CXCR5 | ‡ | GeneTex | GTX100351 | Polyclonal | N/D | AB_1240668 |

*Table 2 continued on next page*

*Table 2 continued*

| Antibody name | Target protein | Performance | Vendor | Catalog no. | Clone | Fluorophore | Research resource Identifier |
|---|---|---|---|---|---|---|---|
| anti-CXCR5 | CXCR5 | ‡ | R and D | MAB-190-SP | 51505 | N/D | AB_2292654 |
| anti-FOXO3a | FOXO3a | ‡ | CST | 2497 | 75D8 | N/D | AB_836876 |
| anti-GZMB | Granzyme B | ‡ | Abcam | AB4059 | Polyclonal | N/D | AB_304251 |
| anti-PD1 | PD-1 | ‡ | Abcam | AB63477 | Polyclonal | N/D | AB_2159165 |
| anti-PD1 | PD-1 | ‡ | ThermoFisher | 14-9985-81 | J43 | N/D | AB_468663 |
| anti-PD1 | PD-1 | ‡ | R and D | AF1021 | Polyclonal | N/D | AB_354541 |
| anti-RFP | RFP | ‡ | ThermoFisher | R10367 | Polyclonal | N/D | AB_2315269 |
| CD11C-BV570 | CD11C | ‡ | BioLegend | 117331 | N418 | BV570 | AB_10900261 |
| CD45-BV785 | CD45 | ‡ | BioLegend | 304047 | HI30 | BV785 | AB_2563128 |
| LY6G-BV570 | LY6G | ‡ | BioLegend | 127629 | 1A8 | BV570 | AB_10899738 |

*Show positive/correct signals in multiple samples/tissues.

†Show positive/correct signals in some but not all samples tested.

‡Show no signal or incorrect signals in most samples tested.

DOI: https://doi.org/10.7554/eLife.31657.011

In the current work, we rely exclusively on commercial antibodies that have previously been validated using IHC or conventional immunofluorescence; when feasible we confirm that staining by t-CyCIF resembles what has previously been reported for IHC staining. This does not constitute a sufficient level of testing or validation for discovery science or clinical studies and the patterns of staining described in this paper should therefore be considered illustrative of the t-CyCIF approach rather than definitive descriptions; we are currently developing a database of matched t-CyCIF and IHC images across multiple tissues and knockdown cell lines to address this issue and share validation test data with the wider research community.

## Fluorophore inactivation, cycle count and tissue integrity

The efficiency of fluorophore inactivation by hydrogen peroxide, light and high pH varies with fluorophore but only minimally with the antibody to which the fluorophore is coupled (Alexa Fluor 488 is inactivated more slowly than Alexa Fluor 570 or 647; *Figure 4B* and *Figure 4—figure supplement 1*). We typically incubate specimens in bleaching conditions for 60 min, which is sufficient to reduce fluorescence intensity by $10^2$ to $10^3$-fold (*Figure 4C*). When testing new antibodies or analyzing new tissues, imaging is performed after each bleaching step and prior to initiation of another t-CyCIF cycle to ensure that fluorophore inactivation is complete. In preliminary studies, we have tested a range of other fluorophores for their compatibility with t-CyCIF including FITC, TRITC, phycoerythrin, Allophycocyanin, eFluor 570 and eFluor 660 (eBioscience). We conclude that it will be feasible to increase the number of t-CyCIF channels per cycle from four to at least six (3 to 5 antibodies plus a DNA stain). However, all the images in this paper are collected using a four-channel method.

The primary limitation on the number of t-CyCIF cycles that can be performed is the integrity of the tissue: some tissues samples are physically more robust and can withstand more staining and washing procedures than others (*Figure 4D*). To study the effect of cycle number on tissue integrity, we performed a 10-cycle t-CyCIF experiment on a tissue microarray (TMA) comprising a total of 40 cores from 16 different tissues and tumor types. After each t-CyCIF cycle, the number of nuclei remaining was quantified for each core relative to the initial number. For example, *Figure 4D* shows breast, bladder, lung and prostate cores in which cell number was reduced after 10 cycles by ~2% and an unusually high 46% (apparent increases in cell number in these data are caused by fluctuation in the performance of cell segmentation routines and are not statistically significant). Cells that were lost appear red in these images. The data show that cell loss is often uneven across samples, preferentially affecting regions of tissue with low cellularity.

Overall, we found that the extent of cell loss varied with tissue type and, within a single tissue type, from core to core (six breast cores are shown; *Figure 4E*). For many tissues, we have not yet

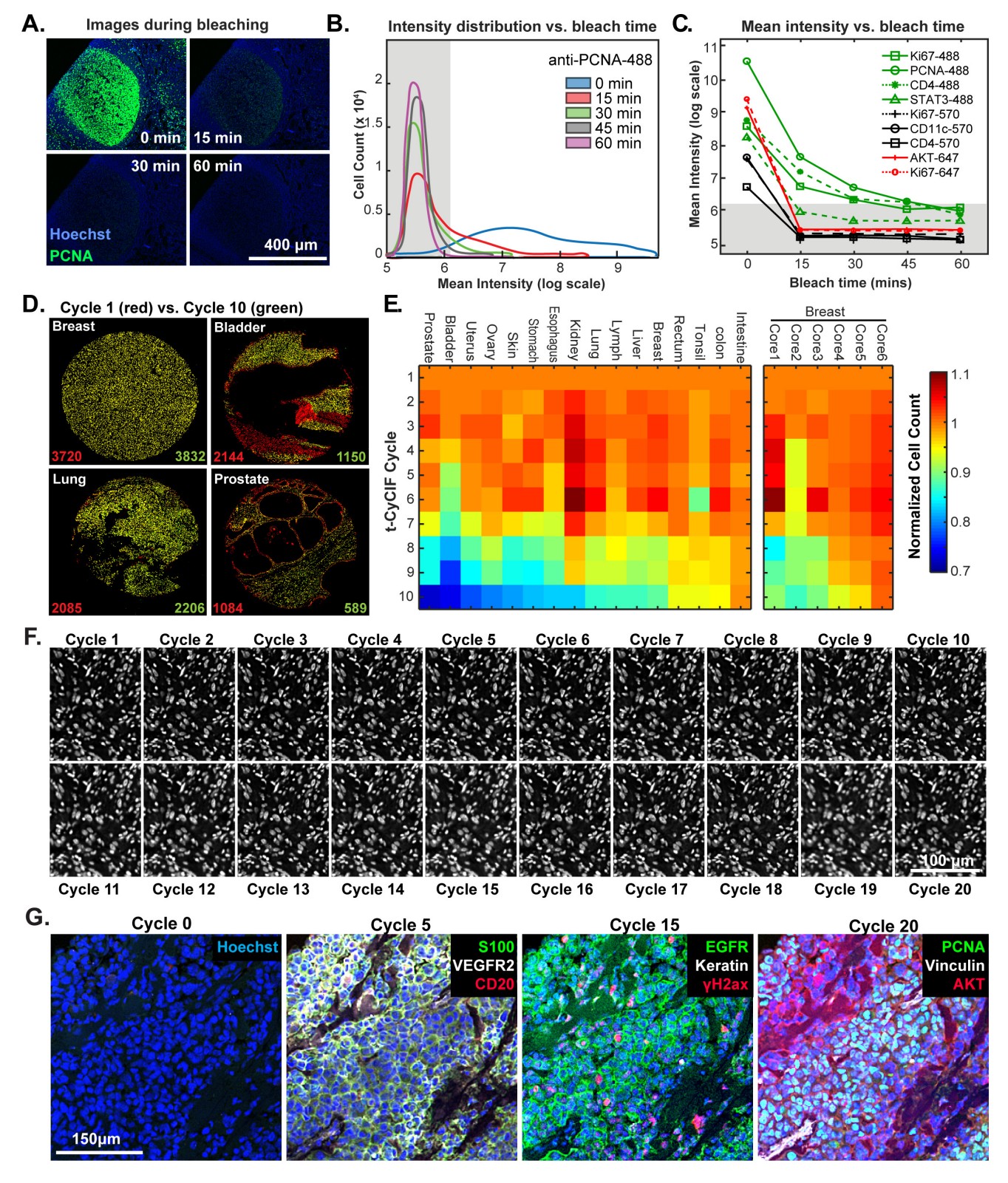

**Figure 4.** Efficacy of fluorophore inactivation and preservation of tissue integrity. (**A**) Exemplary image of a human tonsil stained with PCNA-Alexa 488 that underwent 0, 15, 30 or 60 min of fluorophore inactivation. (**B**) Effect of bleaching duration on the distribution of anti-PCNA-Alexa 488 staining intensities for samples used in (**A**). The distribution is computed from mean values for the fluorescence intensities across all cells in the image that were successfully segmented. The gray band denotes the range of background florescence intensities (below 6.2 in log scale). (**C**) Effect of bleaching

*Figure 4 continued on next page*

*Figure 4 continued*

duration on mean intensity for nine antibodies conjugated to Alexa fluor 488, efluor 570 or Alexa fluor 647. Intensities were determined as in (**B**). The gray band denotes the range of background florescence intensities. (**D**) Impact of t-CyCIF cycle number on tissue integrity for four exemplary tissue cores. Nuclei present in the first cycle are labeled in red and those present after the 10th cycle are in green. The numbers at the bottom of the images represent nuclear counts in cycle 1 (red) and cycle 10 (green), respectively. (**E**) Impact of t-CyCIF cycle number on the integrity of a TMA containing 48 biopsies obtained from 16 different healthy and tumor tissues (see Materials and methods for TMA details) stained with 10 rounds of t-CyCIF. The number of nuclei remaining in each core was computed relative to the starting value; small fluctuations in cell count explain values > 1.0 and arise from errors in image segmentation. Data for six different breast cores is shown to the right. (**F**) Nuclear staining of a melanoma specimen subjected to 20 cycles of t-CyCIF emphasizes the preservation of tissue integrity (22 ± 4%). (**G**) Selected images of the specimen in (**F**) from cycles 0, 5, 15 and 20.
DOI: https://doi.org/10.7554/eLife.31657.012

The following source data and figure supplement are available for figure 4:

**Source data 1.** Mean intensity versus bleach time for multiple antibodies (*Figure 4C*).
DOI: https://doi.org/10.7554/eLife.31657.014
**Source data 2.** Intensity distribution for single cells versus bleach time for one antibody (*Figure 4B*).
DOI: https://doi.org/10.7554/eLife.31657.015
**Source data 3.** Cell counts dependent on number of staining cycles (*Figure 4E*).
DOI: https://doi.org/10.7554/eLife.31657.016
**Figure supplement 1.** Impact of bleaching time on fluorophore inactivation.
DOI: https://doi.org/10.7554/eLife.31657.013

attempted to optimize cycle number and the experiments performed to date do not fully control for pre-analytical variables (*Vassilakopoulou et al., 2015*) such as fixation time and the age of tissue blocks. As a rule, we find that normal tonsil, skin, glioblastoma, ovarian cancer, pancreatic cancer and melanoma can be subjected to >15 cycles with less than 25% cell loss. *Figure 4F* shows a melanoma specimen subjected to 20 t-CyCIF cycles with good preservation of cell and tissue morphology (*Figure 4G*). We conclude that t-CyCIF is compatible with multiple normal tissues and tumor types but that some tissues and/or specimens can be subjected to more cycles than others. One requirement for high cycle number appears to be cellularity: samples in which cells are very sparse tend to be more fragile. We expect improvements in cycle number with additional experimentation and the use of fluidic devices that deliver staining and wash liquids more gently.

One potential concern about cyclic immunofluorescence is that the process is relatively slow; each cycle takes 6–8 hr and we typically perform one cycle per day. However, a single operator can easily process 30 slides in parallel, and in the case of TMAs, 30 slides can comprise over 2000 different samples. Under these conditions, the most time-consuming step in t-CyCIF is collecting the 200–400 fields of view needed to image each slide. Time could be saved by imaging fewer cells per sample, but the results described below (demonstrating substantial cellular heterogeneity in a single piece of a tumor resection) strongly argue in favor of analyzing as large a fraction of each tissue specimen as possible. As a practical matter, data analysis and data interpretation remain more time-consuming than data collection. We also note that the throughput of t-CyCIF compares favorably with other tissue-imaging platforms or single-cell transcriptome profiling.

## Impact of cycle number on immunogenicity

Because t-CyCIF assembles multiplex images sequentially, it is sensitive to factors that alter immunogenicity as cycle number increases. To investigate such effects, we performed a 16-cycle t-CyCIF experiment in which the order of antibody addition was varied between two immediately adjacent tissue slices cut from the same tissue block (*Figure 5A*; Slides A and B); the study was repeated three times, once with tonsil and twice with melanoma specimens with similar results (~1.8 × 10$^5$ cells were used for the analysis and overall cell loss was <15%).

This experiment made it possible to judge: (i) the repeatability of staining a single specimen using the same set of antibodies (*Figure 5A*, denoted by yellow highlight) (ii) the similarity of staining between slides A and B (blue highlight) and (iii) the effect of swapping the order of antibody addition (cycle number) between slides A and B (blue lines). Comparisons within a single slide were made on a cell-by-cell basis but because slides A and B contain different cells, comparisons between slides were made at the level of intensity distributions (computed on a per-cell basis following segmentation). The repeatability of staining (as measured in cycles 3, 7, 12 and 16) was performed using

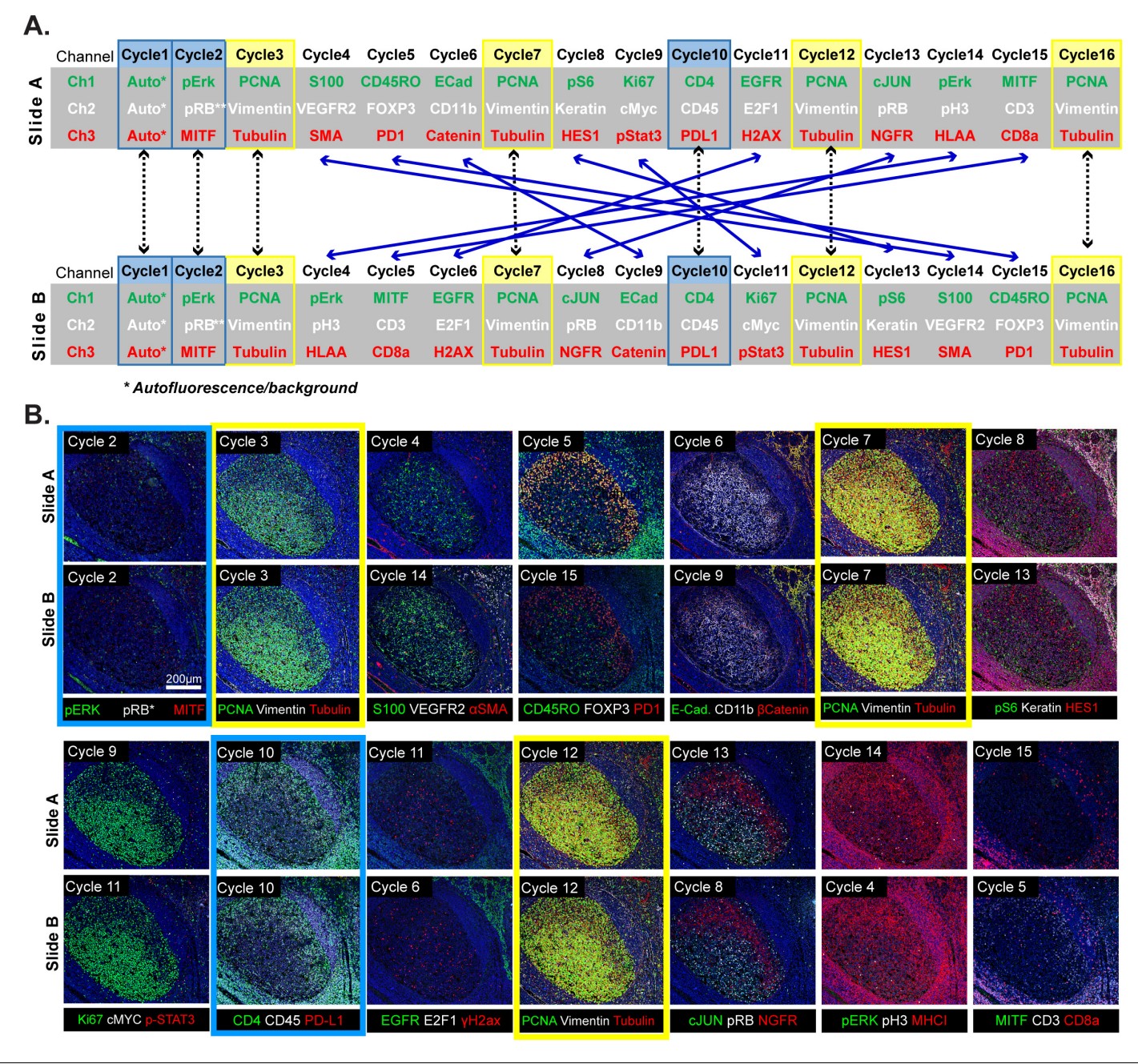

**Figure 5.** Design of a 16-cyle experiment used to assess the reliability of t-CyCIF data. (**A**) t-CyCIF experiment involving two immediately adjacent tissue slices cut from the same block of tonsil tissue (Slide A and Slide B). The antibodies used in each cycle are shown (antibodies are described in *Supplementary file 2*). Highlighted in blue are cycles in which the same antibodies were used on slides A and B at the same time to assess reproducibility. Highlighted in yellow are cycles in which antibodies targeting PCNA, Vimentin and Tubulin were used repeatedly on both slides A and B to assess repeatability. Blue arrows connecting Slides A and B show how antibodies were swapped among cycles. (**B**) Representative images of Slide A (top panels) and Slide B specimens (bottom panels) after each t-CyCIF cycle. The color coding highlighting specific cycles is the same as in A.
DOI: https://doi.org/10.7554/eLife.31657.017

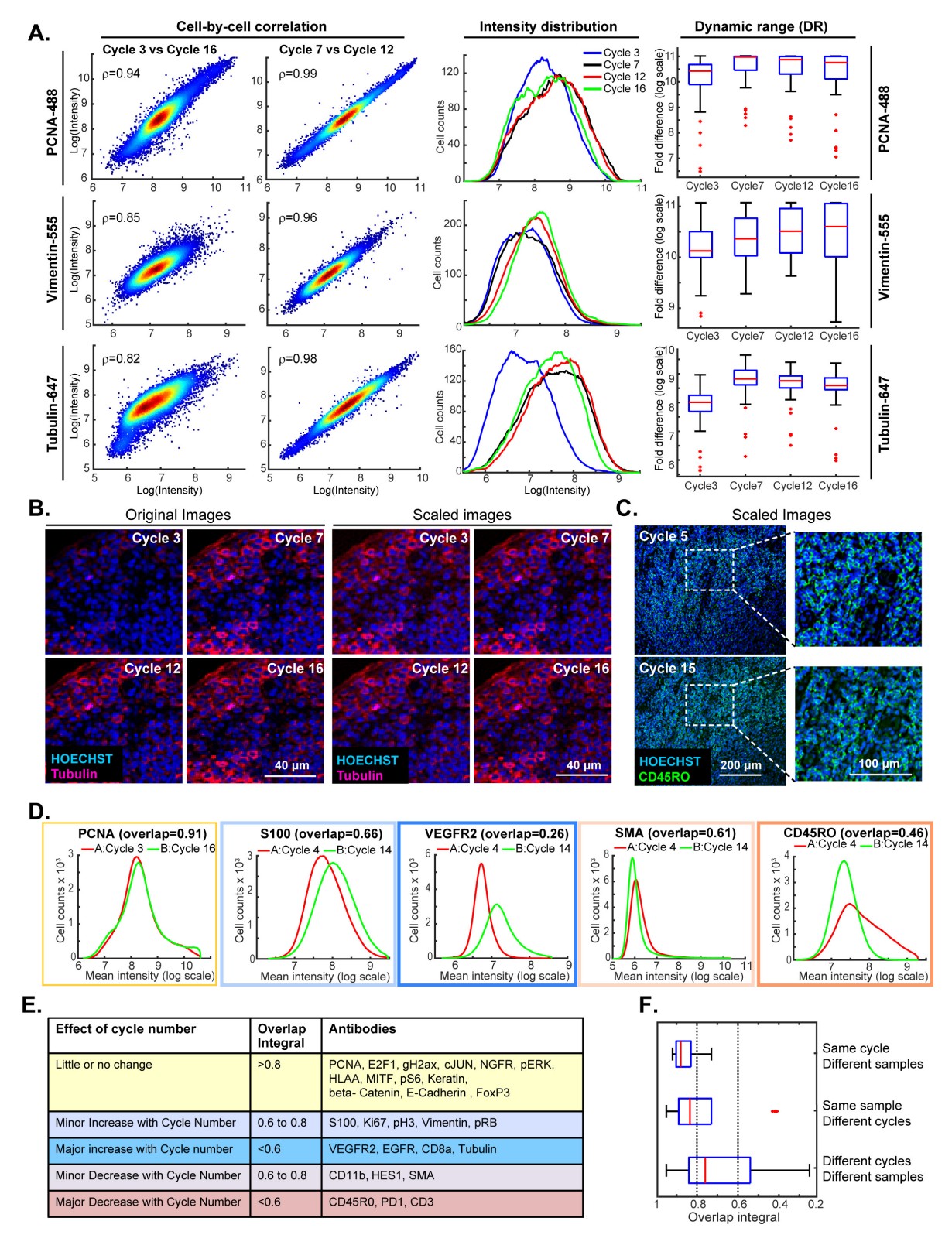

**Figure 6.** Impact of cycle number on repeatability, reproducibility and strength of t-CyCIF immuno-staining. (**A**) Plots on left: comparison of staining intensity for anti-PCNA Alexa 488 (top), anti-vimentin Alexa 555 (middle) and anti-tubulin Alexa 647 (bottom) in cycle 3 vs. 16 and cycle 7 vs. 12 of the 16-cycle t-CyCIF experiment show in *Figure 5*. Intensity values were integrated across whole cells and the comparison is made on a cell-by-cell basis. Spearman's correlation coefficients are shown. Plots in middle: intensity distributions at cycles 3 (blue), 7 (yellow), 12 (red) and 16 (green); intensity

*Figure 6 continued on next page*

*Figure 6 continued*

values were integrated across whole cells to construct the distribution. Box plots to right: estimated dynamic range at four cycle numbers 3, 7, 12, 16. Red lines denote median intensity values (across 56 frames), boxes denote the upper and lower quartiles, whiskers indicate values outside the upper/lower quartile within 1.5 standard deviations, and red dots represent outliers. (B) Representative images showing anti-tubulin Alexa 647 staining at four t-CyCIF cycles; original images are shown on the left (representing the same exposure time and approximately the same illumination) and images scaled by histogram equalization to similar intensity ranges are shown on the right. (C) Image for anti-CD45RO-Alexa 555 at cycles 5 and 15 scaled to similar intensity ranges as described in (B); the dynamic range (DR) of the cycle 15 image is ~3.3 fold lower than that of the Cycle 5 image, but shows similar morphology. (D) Intensity distributions for selected antibodies that were used in different cycles on Slides A and B. Colors denote the degree of concordance between the slides ranging from high (overlap >0.8 in yellow; PCNA), slightly increased or decreased with increasing cycle (overlap 0.6 to 0.8 in light blue or light red; S100 and SMA) or substantially increased or decreased (overlap <0.6 in red or blue; VEGFR2 and CD45RO). (E) Summary of effects of cycle number on antibody staining based on the degree of overlap in intensity distributions (the overlap integral); color coding is the same as in (D). (F) Effect of cycle number and specimen identity on overlap integrals for all antibodies and all cycles assayed. The red line denotes the median intensity value, boxes denote the upper/lower quartiles, and whiskers indicate values outside the upper/lower quartile and within 1.5 standard deviations, and red dots represent outliers. All the numeric data in *Figures 5* and *6* are available in a Jupyter notebook; see Code Availability section of Materials and methods for details.

DOI: https://doi.org/10.7554/eLife.31657.018

The following source data and figure supplement are available for figure 6:

**Source data 1.** Single-cell intensity data used in *Figure 6*.

DOI: https://doi.org/10.7554/eLife.31657.020

**Figure supplement 1.** Comparison of staining intensities across different cycles at a single-cell level.

DOI: https://doi.org/10.7554/eLife.31657.019

anti-PCNA-Alexa 488, anti-Vimentin-Alexa 555 and anti-Tubulin- Alexa 647 which bind abundant proteins with distrinct cellular distributions (*Figure 5B*). Repeated staining of the same antigen is expected to saturate epitopes, but we reasoned that this effect would be less pronounced the more abundant the antigen. For PCNA, the correlation in staining intensities across four cycles was high ($\rho$ = 0.95 to 0.99) and somewhat lower in the case of Vimentin and Tubulin ($\rho$ = 0.80 to 0.95; *Figure 6A*; a more extensive comparison is shown in *Figure 6—figure supplement 1*). When we examined the corresponding images, it was readily apparent that Tubulin, and to a lesser extent Vimentin, stained more intensely in later than in earlier t-CyCIF cycles (see intensity distributions in *Figure 6A* and images in *Figure 6B*). When images were scaled to equalize the intensity range (by histogram equalization), staining patterns were indistinguishable across all cycles and loss of cells or specific subcellular structures was not obviously a factor (*Figure 6B*, left vs right panels and *Figure 6C*). Thus, for at least a subset of antibodies, staining intensity increases rather than decreases with cycle number whereas background fluorescence falls. As a consequence, dynamic range, defined here as the ratio of the least to the most intense 5% of pixels, frequently increases with cycle number (*Figure 6A* and *Figure 6—figure supplement 1*). These effects were reproducible across slides A and B in all three experiments performed.

When we compared staining between slides A and B for the same antibodies and cycle number, the overlap in intensity distributions was high (>0.85), demonstrating good sample to sample reproducibility (*Zhou and Liu, 2012*). The overlap remained high for the majority of antibodies even when they were used in different cycles on slides A and B, but for some antibodies, signal intensity clearly increased or decreased with cycle number (*Figure 6D*; blue and red outlines). In the case of eight antibodies for which the effect of cycle number was greatest (including tubulin, as discussed above), the overlap in intensity distributions was <0.6 as a consequence of both increases and decreases in staining intensity (*Figure 6E*). Overall, we found that the repeatability of staining between two biological samples was highest when the antibodies were used in the same cycle on both samples, lower when the antibodies were used in different cycles on the sample, and lowest when both the order and sample were different (*Figure 6F*).

The reasons for changes in staining intensity with cycle number are not known, but the fact that the same changes were observed across multiple experiments (for any single antibody) suggests that they arise not from irreproducibility of the t-CyCIF procedure but rather from changes in epitope accessibility. Even in these cases, it appears that it is absolute intensity rather than morphology that is variable. Thus, while changes in staining intensity with cycle number are a concern for a subset of t-CyCIF antibodies, it should be possible to minimize the problem by staining all samples in the

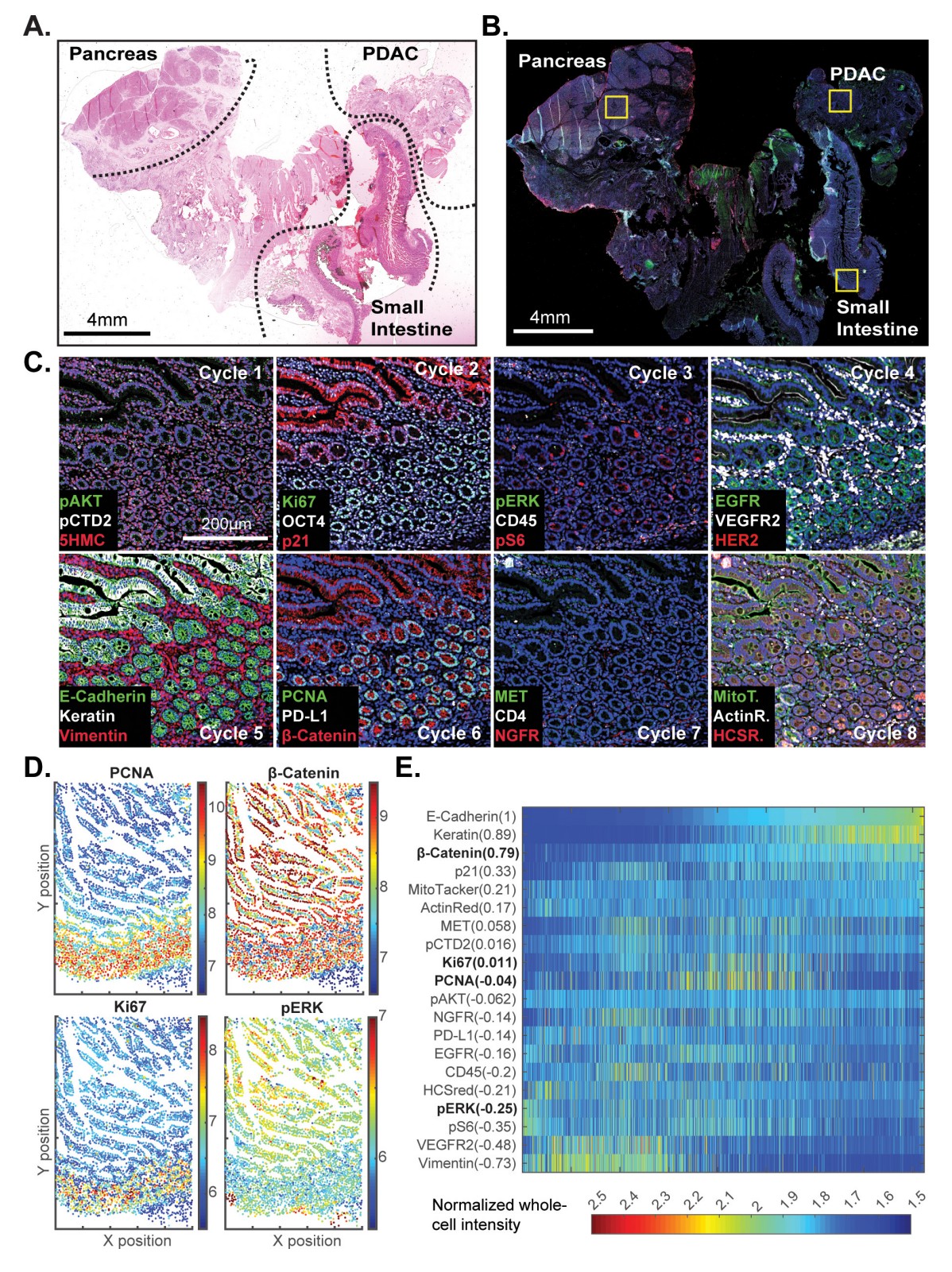

**Figure 7.** t-CyCIF of a large resection specimen from a patient with pancreatic cancer. (A) H&E staining of pancreatic ductal adenocarcinoma (PDAC) resection specimen that includes portions of cancer and non-malignant pancreatic tissue and small intestine. (B) The entire sample comprising 143 stitched 10X fields of view is shown. Fields that were used for downstream analysis are highlighted by yellow boxes. (C) A representative field of normal intestine across 8 t-CyCIF rounds; see *Supplementary file 3* for a list of antibodies. (D) Segmentation data for four antibodies; the color indicates

*Figure 7 continued on next page*

*Figure 7 continued*

fluorescence intensity (blue = low, red = high). (**E**) Quantitative single-cell signal intensities of 24 proteins (rows) measured in ~4×10³ cells (columns) from panel (**C**). The Pearson correlation coefficient for each measured protein with E-cadherin (at a single-cell level) is shown numerically. Known dichotomies are evident such as anti-correlated expression of epithelial (E-Cadherin) and mesenchymal (Vimentin) proteins. Proteins highlighted in red are further analyzed in *Figure 8*.

DOI: https://doi.org/10.7554/eLife.31657.021

The following source data and figure supplement are available for figure 7:

**Source data 1.** Single-cell intensity data used in *Figure 7E*.
DOI: https://doi.org/10.7554/eLife.31657.023
**Source data 2.** Single-cell intensity data used in *Figures 7* and *8*.
DOI: https://doi.org/10.7554/eLife.31657.024
**Figure supplement 1.** t-CyCIF for examining large resection specimens of a human pancreatic cancer.
DOI: https://doi.org/10.7554/eLife.31657.022

same order. Other approaches will also be important; for example, using calibration standards and identifying antibodies exhibiting the least variation with cycle number.

One way to reduce artefacts generated by differences in the order of antibody addition is to create a single high-plex antibody mixture and then stain all antigens in parallel. This approach is not compatible with t-CyCIF but is feasible using methods such as MIBI or CODEX (*Angelo et al., 2014*; *Goltsev, 2017*). However, there is substantial literature showing that the formulation of highly multiplex immuno-assays is complicated by interaction among antibodies (*Ellington et al., 2010*) that has a physicochemical explanation in some cases in weak self-association and viscosity (*Wang et al., 2018*). Consistent with these data, we have observed that when eight or more unlabeled antibodies are added to a t-CyCIF experiment, the intensity of staining can fall, although the effect is smaller than observed with antibodies most sensitive to order of addition. We conclude that the construction of sequentially applied t-CyCIF antibody panels and of single high-plex mixtures will both require optimization of specific panels and their method of use.

## Analysis of large specimens by t-CyCIF

Review of large histopathology specimens by pathologists involves rapid and seamless switching between low-power fields to scan across large regions of tissue and high-power fields to study cellular morphology. To mimic this integration of information at both tissue and cellular scales, we performed eight-cycle t-CyCIF on a large 2 × 1.5 cm resection specimen that includes pancreatic ductal adenocarcinoma (PDAC) and adjacent normal pancreatic tissue and small intestine (*Figure 7A–C*). Nuclei were located in the DAPI channel and cell segmentation performed using a watershed algorithm (*Figure 7—figure supplement 1*: see Materials and methods section for a discussion of the method and its caveats) yielding ~2 × 10⁵ single cells each associated with a vector comprising 25 whole-cell fluorescence intensities. Differences in subcellular distribution were evident for many proteins, but for simplicity, we only analyzed fluorescence intensity on a per-antigen basis integrated over each whole cell. Results were visualized by plotting intensity value onto the segmentation data (*Figure 7D*), by computing correlations on a cell-by-cell basis (*Figure 7E*), or by using t-distributed stochastic neighbor embedding (t-SNE) (*Maaten and Hinton, 2008*), which clusters cells in 2D based on their proximity in the 25-dimensional space of image intensity data (*Figure 8A*).

The analysis in *Figure 7E* shows that E-cadherin, keratin and β-catenin levels are highly correlated with each other, whereas vimentin and VEGFR2 receptor levels are anti-correlated, recapitulating the known dichotomy between epithelial and mesenchymal cell states in normal and diseased tissues. Many other physiologically relevant correlations are also observed, for example between the levels of pERK$^{T202/Y204}$ (the phosphorylated, active form of the kinase) and activating phosphorylation of the downstream kinase pS6$^{S235/S236}$ ($r = 0.81$). When t-SNE was applied to all cells in the specimen, we found that those identified during histopathology review as being from non-neoplastic pancreas (red) were distinct from PDAC (green) and also from the neighboring non-neoplastic small intestine (blue) (*Figure 8B–D*). Vimentin and E-Cadherin had very different levels of expression in PDAC and normal pancreas as a consequence of epithelial-to-mesenchymal transitions (EMT) in malignant tissues as well as the presence of a dense tumor stroma, a desmoplastic reaction that is a hallmark of the PDAC microenvironment (*Mahadevan and Von Hoff, 2007*). The microenvironment

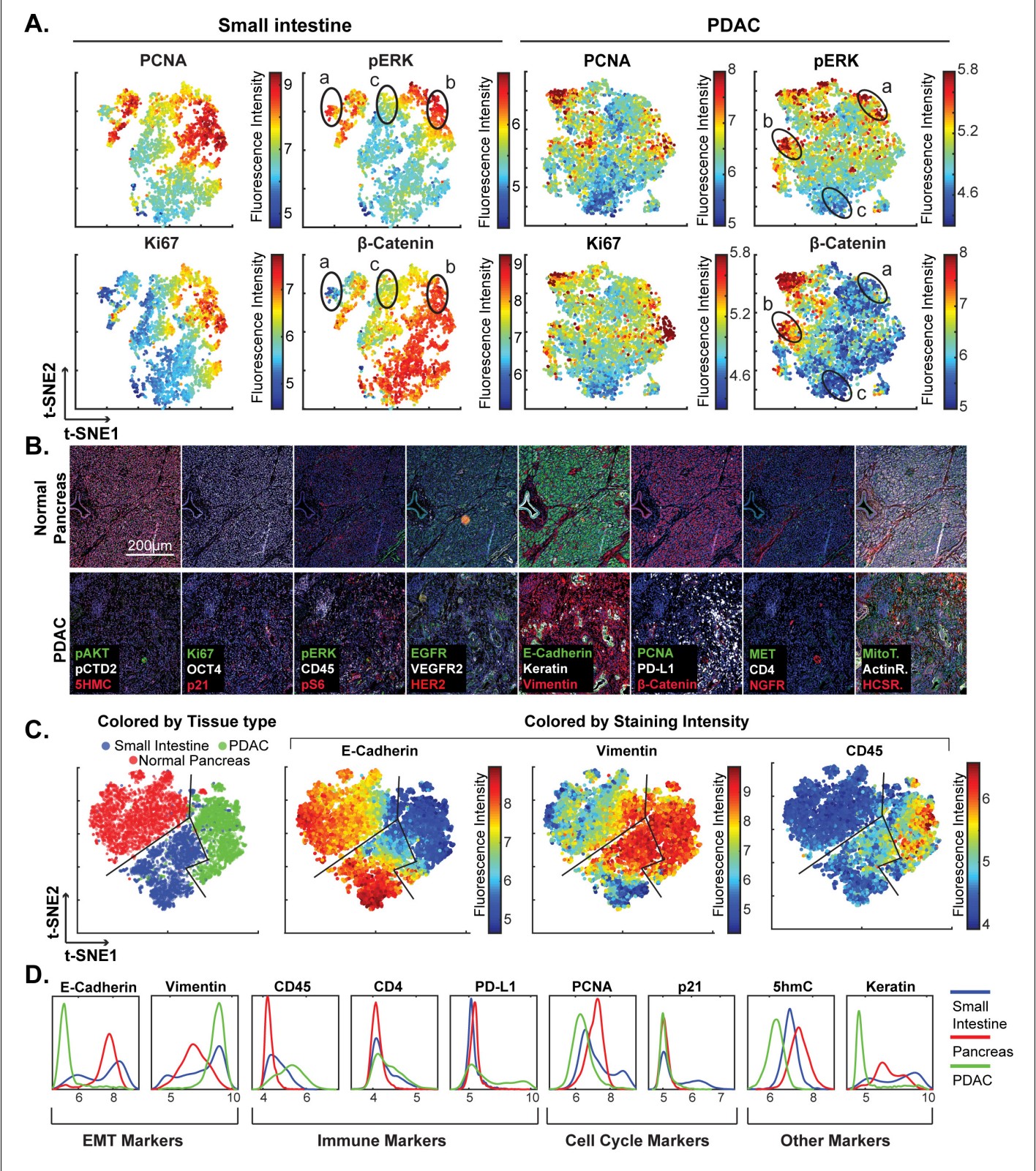

**Figure 8.** High-dimensional single-cell analysis of human pancreatic cancer sample with t-CyCIF. (**A**) t-SNE plots of cells derived from small intestine (left) or the PDAC region (right) of the specimen shown in *Figure 7* with the fluorescence intensities for markers of proliferation (PCNA and Ki67) and signaling (pERK and β-catenin) overlaid on the plots as heat maps. In both tissue types, there exists substantial heterogeneity: circled areas indicate the relationship between pERK and β-catenin levels in cells and represent positive ('a'), negative ('b') or no association ('c') between these markers. (**B**)
*Figure 8 continued on next page*

*Figure 8 continued*

Representative frames of normal pancreas and pancreatic ductal adenocarcinoma from the 8-cycle t-CyCIF staining of the same resection specimen from *Figure 7*. (C) t-SNE representation and clustering of single cells from normal pancreatic tissue (red), small intestine (blue) and pancreatic cancer (green). Projected onto the origin of each cell in t-SNE space are intensity measures for selected markers demonstrating distinct staining patterns. (D) Fluorescence intensity distributions for selected markers in small intestine, pancreas and PDAC.

DOI: https://doi.org/10.7554/eLife.31657.025

The following source data is available for figure 8:

**Source data 1.** Single-cell data in FCS format (*Figure 8C–E*).

DOI: https://doi.org/10.7554/eLife.31657.026

of PDAC was more heavily infiltrated with CD45$^+$ immune cells than the normal pancreas, and the intestinal mucosa of the small intestine was also replete with immune cells, consistent with the known architecture and organization of this tissue.

The capacity to image samples that are several square centimeters in area with t-CyCIF can facilitate the detection of signaling biomarker heterogeneity. The WNT pathway is frequently activated in PDAC and is important for oncogenic transformation of gastrointestinal tumours (*Jones et al., 2008*). Approximately 90% of sporadic PDACs also harbor driver mutations in KRAS, activating the MAPK pathway and promoting tumourigenesis (*Vogelstein et al., 2013*). Studies comparing these pathways have come to different conclusions with respect to their relationship: some studies show concordant activation of MAPK and WNT signaling and others argue for exclusive activation of one pathway or the other (*Jeong et al., 2012*). In t-SNE plots derived from images of PDAC, multiple sub-populations of cells representing negative, positive or no correlation between pERK and β-catenin levels can be seen (marked with labels 'a', 'b' or 'c', respectively in *Figure 8A*). The same three relationships can be found in non-neoplastic pancreas and small intestine (*Figures 8A* and *7C*). In PDAC, malignant cells can be distinguished from stromal cells, to a first approximation, by high proliferative index, which can be measured by staining for Ki-67 and PCNA (*Bologna-Molina et al., 2013*). When we gated for cells that were both Ki67$^{high}$ and PCNA$^{high}$, and thus likely to be malignant, the co-occurrence of different relationship between pERK and β-catenin levels on a cellular level was again evident. While we cannot exclude the possibility of phospho-epitope loss during sample preparation, it appears that the full range of possible relationships between the MAPK and WNT signaling pathways described in the literature can be found within a specimen from a single patient, illustrating the impact of tissue context on the activities of key signal transduction pathways.

## Multiplex imaging of immune infiltration

Immuno-oncology drugs, including immune checkpoint inhibitors targeting CTLA-4 and the PD-1/PD-L1 axis are rapidly changing the therapeutic possibilities for traditionally difficult-to-treat cancers including melanoma, renal and lung cancers, but responses are variable across and within cancer types. The hope is that tumor immuno-profiling will yield biomarkers predictive of therapeutic response in individual patients. For example, expression of PD-L1 correlates with responsiveness to the ICIs pembrolizumab and nivolumab (*Mahoney and Atkins, 2014*) but the negative predictive value of PD-L1 expression alone is insufficient to stratify patient populations (*Sharma and Allison, 2015*). In contrast, by measuring PD-1, PD-L1, CD4 and CD8 by IHC on sequential tumor slices, it has been possible to identify some immune checkpoint inhibitor-responsive melanom patients (*Tumeh et al., 2014*). To test t-CyCIF in this application, eight-cycle imaging was performed on a 1 × 2 cm specimen of clear-cell renal cell carcinoma using 10 antibodies against multiple immune markers and 12 against other proteins expressed in tumor and stromal cells (*Figure 9A–B*; *Supplementary file 4*). A region of the specimen corresponding to tumor was readily distinguishable from non-malignant stroma based on α-SMA expression (α-SMA$^{high}$ regions denote stroma and α-SMA$^{low}$ regions high density of malignant cells).

In the α-SMA$^{low}$ domain, CD3$^+$ or CD8$^+$ lymphocytes were fourfold enriched (*Figure 9C*) and PD-1 and PD-L1-positive cells were 13 to 20-fold more prevalent as compared to the surrounding tumor stroma (α-SMA$^{high}$ domain); CD3$^+$ CD8$^+$ double positive T-cells were found almost exclusively in the tumor. Suppression of immune cells is mediated by binding of PD-L1 ligand, which is commonly expressed by tumor cells, to the PD1 receptor expressed on immune cells (*Tumeh et al., 2014*). To

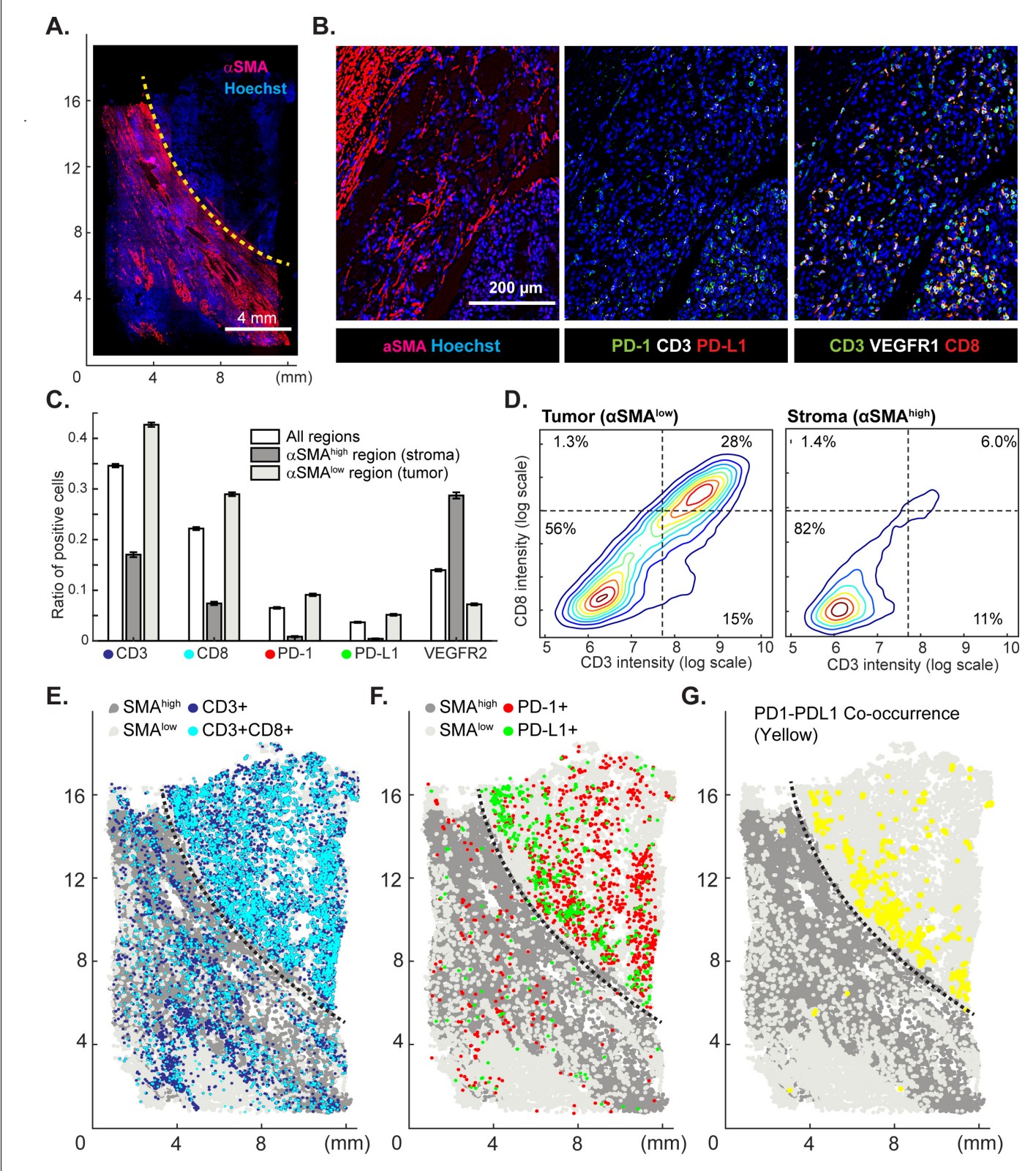

**Figure 9.** Spatial distribution of immune infiltrates and checkpoint proteins. (**A**) Low-magnification image of a clear cell renal cancer subjected to 12-cycle t-CyCIF (see *Supplementary file 4* for a list of antibodies). Regions high in α-smooth muscle actin (α-SMA) correspond to stromal components of the tumor, those low in α-SMA represent regions enriched for malignant cells. (**B**) Representative images from selected t-CyCIF channels are shown. (**C**) Quantitative assessment of total lymphocytic cell infiltrates (CD3[+] cells), CD8[+] T lymphocytes, cells expressing PD-1 or its ligand PD-L1 or the VEGFR2

*Figure 9 continued on next page*

*Figure 9 continued*

for the entire tumor or for α-SMA[high] and α-SMA[low] regions. VEGFR2 is a protein primarily expressed in endothelial cells and is targeted in the treatment of renal cell cancer. The error bars represent the S.E.M. derived from 100 rounds of bootstrapping. (D) Density plot for CD3 and CD8 expression on single cells in the tumor (left) or stromal domains (right). (E) Centroids of CD3[+] or CD3[+]CD8[+] cells in blue or dark blue as well as cells staining as SMA[high] or SMA[low] (gray and light-gray, respectively) used to define the stromal and tumor regions. (F) Centroids of PD-1[+] and PD-L1[+] cells are shown in red and green, respectively. (G) Results of a K-nearest neighbor algorithm used to compute areas in which PD-1[+] and PD-L1[+] cells lie within ~10 µm of each other and with high spatial density (in yellow) and thus, are potentially positioned to interact at a molecular level.

DOI: https://doi.org/10.7554/eLife.31657.027

The following source data and figure supplement are available for figure 9:

**Source data 1.** Immune cell counts from bootstrapping in tumor and stroma regions (*Figure 9C*).

DOI: https://doi.org/10.7554/eLife.31657.029

**Source data 2.** Single-cell intensity data used in *Figure 9*.

DOI: https://doi.org/10.7554/eLife.31657.030

**Figure supplement 1.** Spatial analysis of PD-1 and PD-L1 expressing cells.

DOI: https://doi.org/10.7554/eLife.31657.028

begin to estimate the likelihood of ligand-receptor interactions, we quantified the degree of co-localization of cells expressing the two molecules. The centroids of PD-1[+] or PD-L1[+] cells were determined from images (PD-1, red; PD-L1, green, *Figure 9E*) and co-localization (highlighted in yellow, *Figure 9F*) computed by k-nearest neighbor analysis. We found that co-localization of PD-1/PD-L1 was ~2.7-fold more likely (*Figure 9—figure supplement 1*) in tumor and stroma and was concentrated on the tumor-stroma border consistent with previous reports on melanoma (*Tumeh et al., 2014*). These data demonstrate the potential of spatially resolved immuno-phenotyping to quantify state and location of tumor infiltrating lymphocytes; such data may ultimately yield biomarkers predictive of sensitivity to immune checkpoint inhibitor (*Tumeh et al., 2014*).

## Analysis of diverse tumor types and grades using t-CyCIF of tissue-microarrays (TMA)

To explore the general utility of t-CyCIF in a range of healthy and cancer tissues we applied eight cycle t-CyCIF to TMAs containing 39 different biopsies from 13 healthy tissues and 26 biopsies corresponding to low- and high-grade cancers from the same tissue types (*Figure 10A* and *Figure 10—figure supplement 1*, *Supplementary file 3* for antibodies used, *Supplementary file 5* for TMA details and naming conventions) and then performed t-SNE and clustering on single-cell intensity data (*Figure 10B*). The great majority of TMA samples mapped to one or a few discrete locations in the t-SNE projection (compare normal kidney tissue - KI1, low-grade tumors - KI2, and high-grade tumors – KI3; *Figure 10C*), although ovarian cancers were scattered across the t-SNE projection (*Figure 10D*); overall, there was no separation between normal tissue and tumors regardless of grade (*Figure 10E*). In a number of cases, high-grade cancers from multiple different tissues of origin co-clustered, implying that transformed morphologies and cell states were closely related. For example, while healthy and low-grade pancreatic and stomach cancer occupied distinct t-SNE domains, high-grade pancreatic and stomach cancers were intermingled and could not be readily distinguished (*Figure 10F*), recapitulating the known difficulty in distinguishing high-grade gastrointestinal tumors of diverse origin by histophathology (*Varadhachary and Raber, 2014*). Nonetheless, t-CyCIF might represent a means to identify discriminating biomarkers by efficiently sorting through large numbers of alternative antigens and antigen localizations.

## Quantitative analysis reveals global and regional heterogeneity and multiple histologic subtypes within the same tumor in glioblastoma multiforme (GBM)

Data from single-cell genomics reveals extensive heterogeneity in many types of cancer (*Turner and Reis-Filho, 2012*) but our understanding of this phenomenon requires spatially resolved data (*Giesen et al., 2014*). We performed eight-cycle imaging on a 2.5 cm x 1.8 mm resected glioblastoma (GBM) specimen imaging markers of neural development, cell cycle state and signal transduction (*Figure 11A–B*, *Supplementary file 6*). GBM is a highly aggressive and genetically heterogeneous (*Brennan et al., 2013*) brain cancer commonly classified into four histologic subtypes

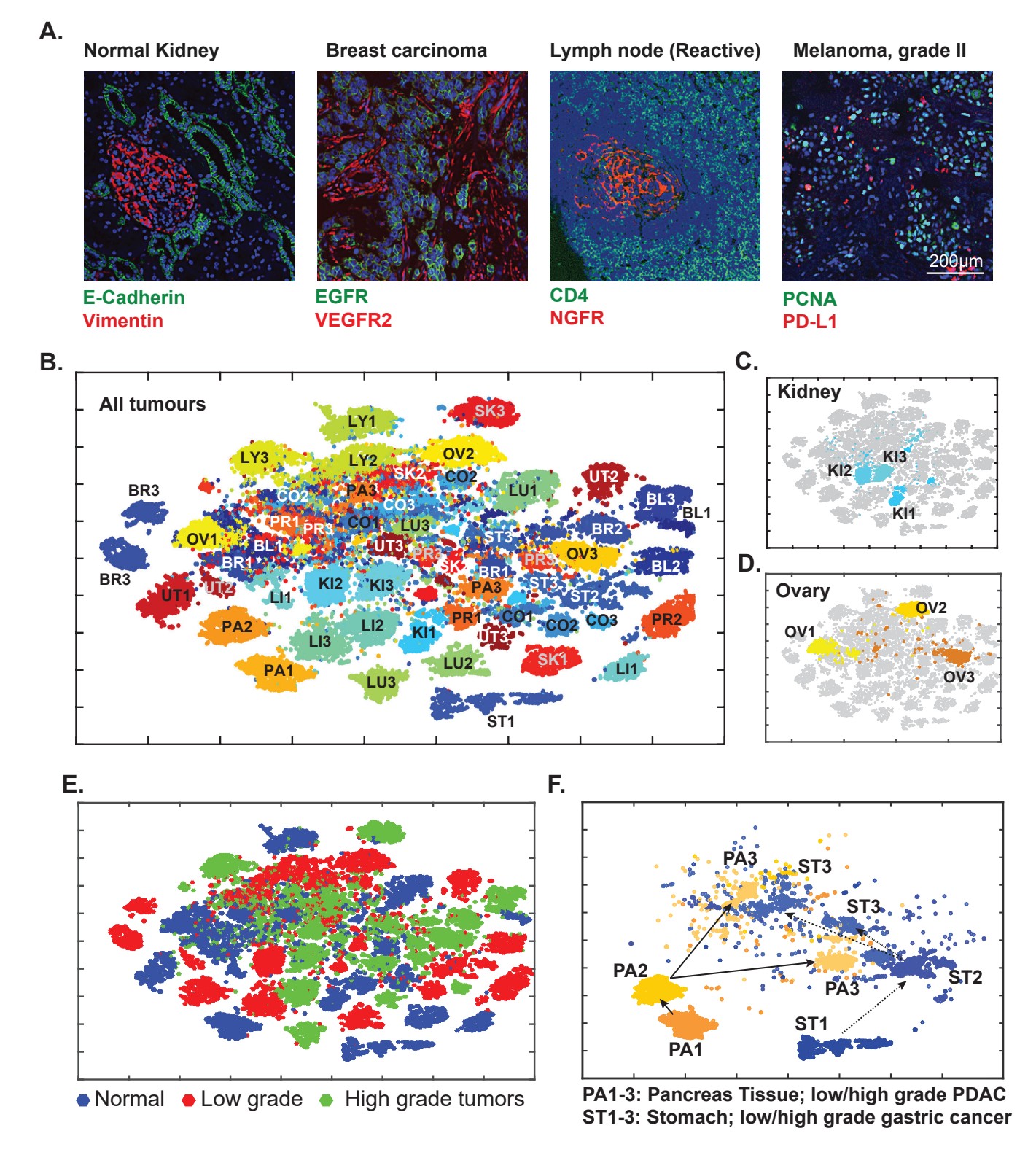

**Figure 10.** Eight-cycle t-CyCIF of a tissue microarray (TMA) including 13 normal tissues and corresponding tumor types. The TMA includes normal tissue types, and corresponding high- and low-grade tumors, for a total of 39 specimens (see *Supplementary file 3* for antibodies and *Supplementary file 5* for specifications of the TMA). (A) Selected images of different tissues illustrating the quality of t-CyCIF images (additional examples shown in *Figure 9—figure supplement 1*; full data available online at www.cycif.org). (B) t-SNE plot of single-cell intensities of all 39 cores;
*Figure 10 continued on next page*

*Figure 10 continued*

data were analyzed using the CYT package (see Materials and methods). Tissues of origin and corresponding malignant lesions were labeled as follows: BL, bladder cancer; BR, breast cancer CO, Colorectal adenocarcinoma, KI, clear cell renal cancer, LI, hepatocellular carcinoma, LU, lung adenocarcinoma, LY, lymphoma, OV, high-grade serous adenocarcinoma of the ovary, PA, pancreatic ductal adenocarcinoma, PR, prostate adenocarcinoma, UT, uterine cancer, SK, skin cancer (melanoma), ST, stomach (gastric) cancer. Numbers refer to sample type; '1' to normal tissue, '2' to -grade tumors and '3' to high-grade tumors. (C) Detail from panel B of normal kidney tissue (KI1) a low-grade tumor (KI2) and a high-grade tumor (KI3) (D) Detail from panel B of normal ovary (OV1) low-grade tumor (OV2) and high-grade tumor (OV3). (E) t-SNE plot from Panel B coded to show the distributions of all normal, low-grade and high-grade tumors. (F) tSNE clustering of normal pancreas (PA1) and pancreatic cancers (low-grade, PA2, and high-grade, PA3) and normal stomach (ST1) and gastric cancers (ST2 and ST3, respectively) showing intermingling of high-grade cells.
DOI: https://doi.org/10.7554/eLife.31657.031

The following source data and figure supplement are available for figure 10:

**Source data 1.** Single-cell intensity data used in *Figure 10*.
DOI: https://doi.org/10.7554/eLife.31657.033
**Figure supplement 1.** Gallery of exemplary tissues imaged on the TMA described in *Figure 10*.
DOI: https://doi.org/10.7554/eLife.31657.032

---

(*Olar and Aldape, 2014*). Following image segmentation, phenotypic heterogeneity was assessed at three spatial scales corresponding to: (i) $1.6 \times 1.4$ mm fields of view (252 total) each of which comprised $10^3$ to $10^4$ cells (ii) seven macroscopic regions of ~$10^4$ to $10^5$ cells each, corresponding roughly to tumor lobes and (iii) the whole tumor comprising ~$10^6$ cells. To quantify local heterogeneity, we computed the informational entropy on a-per-channel basis for $10^3$ randomly selected cells in each field (*Figure 11C*; see online Materials and methods for details). In this setting, informational entropy is a measure of cell-to-cell heterogeneity on a mesoscale corresponding to 10–30 cell diameters. For a marker such as EGFR, which can function as a driving oncogene in GBM, informational entropy was high in some areas (*Figure 11C*; red dots) and low in others (blue dots). Areas with high entropy in EGFR abundance did not co-correlate with areas that were most variable with respect to a downstream signaling protein such as pERK. Thus, the extent of local heterogeneity varied with the region of the tumor and the marker being assayed.

Semi-supervised clustering using expectation–maximization Gaussian mixture (EMGM) modeling of all cells in the tumor yielded eight distinct clusters, four of which encompassed 85% of all cells (*Figure 12A* and *Figure 12—figure supplement 1*). Among these, cluster one had high EGFR levels, cluster two had high NGFR and Ki67 levels and cluster six had high levels of vimentin; cluster five was characterized by high keratin and pERK levels. The presence of four highly populated t-CyCIF clusters is consistent with data from single-cell RNA-sequencing of ~400 cells from five GBMs (*Patel et al., 2014*). Three of the t-CyCIF clusters have properties reminiscent of established histological subtypes including: classical, cluster 1; pro-neural, cluster 3; and mesenchymal, cluster 6, but additional work will be required to confirm such assignments.

To study the relationship between phenotypic diversity and tumor architecture, we mapped each cell to an EMGM cluster (denoted by color). Extensive intermixing was observed at all spatial scales (*Figure 12B*). For example, field of view 147 was highly enriched for cells corresponding to cluster 5 (yellow), but a higher magnification view revealed extensive intermixing of four other cluster types on a scale of ~3–5 cell diameters (*Figure 12C*). At the level of larger, macroscopic tumor regions, the fraction of cells from each cluster also varied dramatically (*Figure 12D*). None of these findings was substantially different when the number of clusters was set to 12 (*Figure 12—figure supplement 2*).

These results have several implications. First, they suggest that GBM is phenotypically heterogeneous on a spatial scale of 5–1000 cell diameters and that cells corresponding to distinct t-CyCIF clusters are often found in the vicinity of each other. Second, sampling a small region of a large tumor has the potential to misrepresent the proportion and distribution of tumor subtypes, with implications for prognosis and therapy. Similar concepts likely apply to other tumor types with high genetic heterogeneity, such as metastatic melanoma (*Tirosh et al., 2016*), and are therefore relevant to diagnostic and therapeutic challenges arising from tumor heterogeneity.

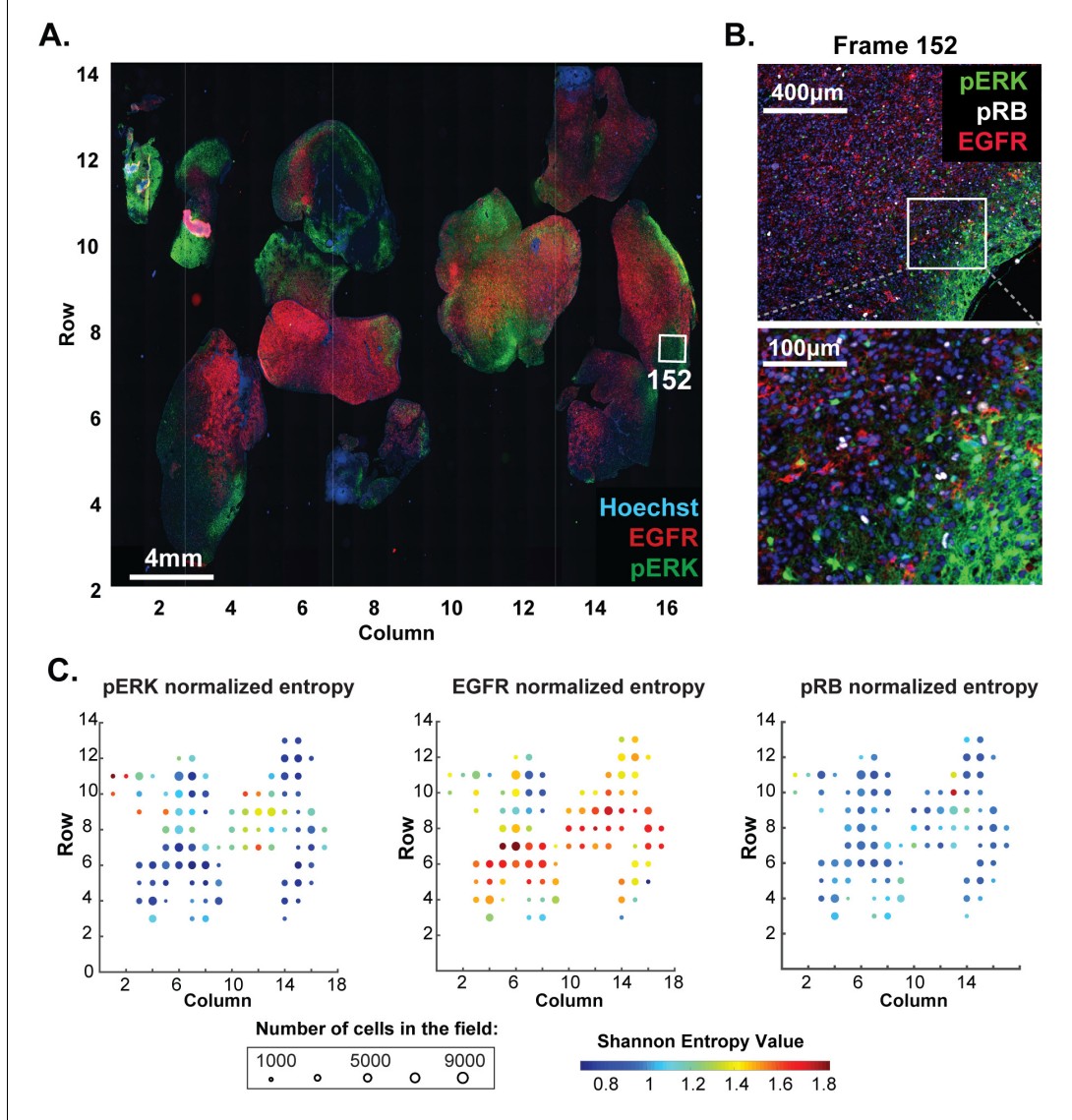

**Figure 11.** Molecular heterogeneity in a single GBM tumor. (**A**) Representative low-magnification image of a GBM specimen generated from 221 stitched 10X frames; the sample was subjected to 10 rounds of t-CyCIF using antibodies listed in *Supplementary file 6*. (**B**) Magnification of frame 152 (whose position is marked with a white box in panel A) showing staining of pERK, pRB and EGFR; lower panel shows a further magnification to allow single cells to be identified. (**C**) Normalized Shannon entropy of each of 221 fields of view to determine the extent of variability in signal intensity for 1000 cells randomly selected from that field for each of the antibodies shown. The size of the circles denotes the number of cells in the field and the color represents the value of the normalized Shannon entropy (data are shown only for those fields with more than 1000 cells; see Materials and methods for details).

DOI: https://doi.org/10.7554/eLife.31657.034

The following source data is available for figure 11:

**Source data 1.** Normalized entropy data shown in *Figure 11C*.
DOI: https://doi.org/10.7554/eLife.31657.035
**Source data 2.** Single-cell intensity data used in *Figure 11* and *12*.
DOI: https://doi.org/10.7554/eLife.31657.036

## Discussion

The complex molecular biology and spatial organization of tissues and solid tumors poses a scientific and diagnostic challenge that is not sufficiently addressed using single-cell genomics, in which morphology is commonly lost, or H&E and single-channel IHC staining, which provide data on only a few

proteins or molecular features. At the same time, the vast number of FFPE histological specimens collected in the course of routine clinical care and clinical trials (and in the study of model organisms) represents an underutilized resource with great potential for novel discovery. A variety of methods for performing highly multiplexed immune-based imaging of cells and tissues has recently been described including imaging cytometry (*Giesen et al., 2014*), MIBI (*Angelo et al., 2014*), DNA-exchange imaging (DEI) (*Wang, 2017*) and CODEX (*Goltsev, 2017*); FISSEQ (*Lee et al., 2014*) directly images expressed RNAs. Like traditional antibody stripping approaches, the cyclic immuno-fluorescence approach first described by Gerdes et al (*Gerdes et al., 2013*) and further developed here assembles highly multiplexed images by sequential acquisition of lower dimensional immunoflu-orescence images. We show here that the t-CyCIF implementation of cyclic immunofluorescence is compatible with a wide range of antibodies and tissue types and yields up to 60-plex images with excellent preservation of small intracellular structures.

The requirement in t-CyCIF for multiple rounds of staining and imaging might seem to be a liability but it has several substantial advantages relative to all-in-one methods such as MIBI, DEI and CODEX. First, t-CyCIF can be performed using existing fluorescence microscopes. Not only does this reduce costs and barriers to entry, it allows the unique strengths of slide-scanning, confocal, and structured illumination microscopes to be exploited. Using different instruments, samples several square centimeters in area can be rapidly analyzed at resolutions of ~1 μm and selected fields of view studied at super-resolution (~110 nm on an OMX Blaze). Multiscale imaging makes it possible to combine tissue-level architecture with subcellular morphology, much like a pathologist switching between low- and high-power fields, but there is little chance that such capabilities can be combined in a single instrument. Because no spectral deconvolution is required, t-CyCIF can use highly optimized filter sets and fluorophores, resulting in good sensitivity. t-CyCIF antibody panels are also simple to assemble and validate using commercial antibodies, including those that constitute FDA-approved diagnostics. This avoids the limitations of an exlusive reliance on pre-assembled reagent kits provided by manufacturers. Finally, t-CyCIF is compatible with H&E staining, enabling fluorescence imaging to be combined with conventional histopathology review.

Commercial systems for non-optical tissue imaging are only now starting to appear and it is difficult to compare their performance to multiplexed immunofluorescence, particularly because the approach published by *Gerdes et al. (2013)* is proprietary and available only as commercial service. In contrast, the t-CyCIF method described here can easily be implemented in a conventional research or clinical laboratory without the need for expensive equipment or specialized reagents. As MIBI, DEI and CODEX instruments come on-line, direct comparison with t-CyCIF will be possible. We anticipate that high resolution and good linearity will be areas in which fluorescence imaging is superior to enzymatic amplification, laser ablation or mechanical picking of tissues. t-CyCIF is relatively slow when performed on a single sample, but when many large specimens or TMAs are processed in parallel, throughput is limited primarily by imaging acquisition, which is at least as fast as approaches involving laser ablation. Considerable opportunity exists for further improvement in t-CyCIF by switching from four to six-channels per cycle, optimizing bleach and processing solutions to preserve tissue integrity, using fluidic devices to rapidly process many slides in parallel and developing better software for identifying fields of view that can be skipped in large irregular specimens. Because direct fluorescence will remain challenging in the case of very rare epitopes, we speculate that hybrid approaches involving t-CyCIF and methods such as DEI or CODEX will ultimately prove to be most effective.

As in all methods involving immune detection, antibodies are the most critical and difficult to validate reagents in t-CyCIF. To date, we have shown that over 200 commercial antibodies are compatible with the method as judged by patterns of staining similar to those previously reported for IHC; this is an insufficient level of validation for most studies and we are therefore working to develop a generally useful antibody validation resource (www.cycif.org). Thus, while this paper describes markers relevant to diagnosis of disease, our results are illustrative of the t-CyCIF approach and specific findings might not prove statistically significant when tested on larger, well-controlled sets of human samples.

There is little or no evidence that antigenicity falls across the board in t-CyCIF as cycle number increases; signal-to-noise ratios can even increase due to falling background auto-fluorescence. When samples are stained with the same antibodies in different t-CyCIF cycles, repeatability is high (as measured by correlation in staining intensity on a cell-by-cell basis) as is reproducibility across

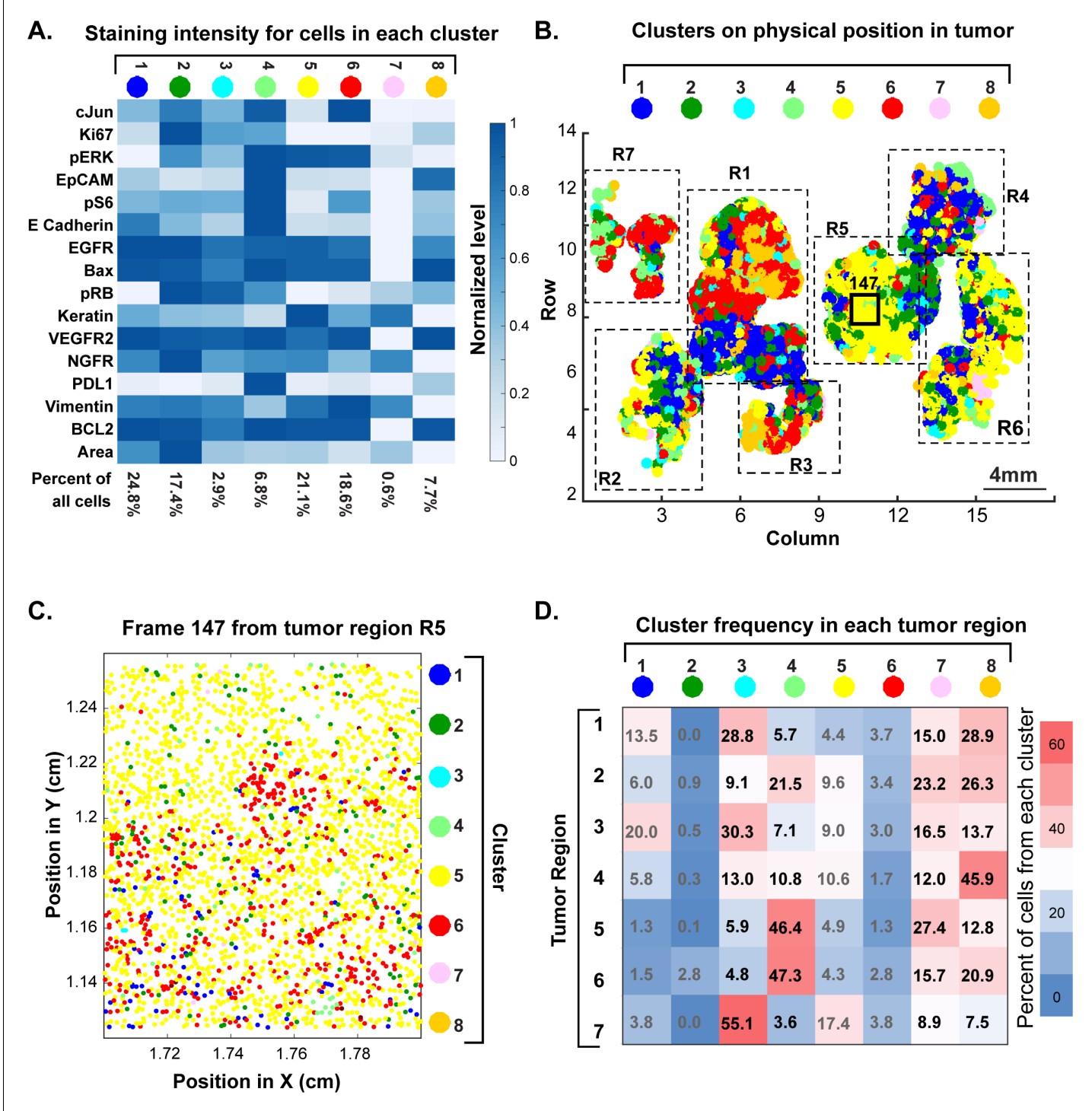

**Figure 12.** Spatial distribution of molecular phenotypes in a single GBM. (A) Clustering of intensity values for 30 antibodies in a 10-cycle t-CyCIF analysis integrated over each whole cell based on images shown in *Figure 11*. Intensity values were clustered using expected-maximization with Gaussian mixtures (EMGM), yielding eight clusters, of which four clusters accounted for the majority of cells. The intensity scale shows the average level for each intensity feature in that cluster. The number of cells in the cluster is shown as a percentage of all cells in the tumor (bottom of panel). An analogous analysis is shown for 12 clusters in *Figure 12—figure supplement 2*. (B) EMGM clusters (in color code) mapped back to the positions of individual cells in the tumor. The coordinate system is the same as in *Figure 11A*. The positions of seven macroscopic regions (R1-R7) representing distinct lobes of the tumor are also shown. (C) Magnified view of Frame 147 from region R5 with EMGM cluster assignment for each cell in the frame; dots represent the centroids of single cells. (D) The proportional representation of EMGM clusters in each tumor region as defined in panel (B).
DOI: https://doi.org/10.7554/eLife.31657.037

*Figure 12 continued on next page*

*Figure 12 continued*

The following source data and figure supplements are available for figure 12:

**Source data 1.** Ratios of EMGM clusters in different regions of a GBM (*Figure 12D*).
DOI: https://doi.org/10.7554/eLife.31657.040
**Figure supplement 1.** Determination of cluster number for semi-supervised clustering using expectation–maximization Gaussian mixture (EMGM) modeling.
DOI: https://doi.org/10.7554/eLife.31657.038
**Figure supplement 2.** Spatial distribution of molecular phenotypes in a single GBM.
DOI: https://doi.org/10.7554/eLife.31657.039

two successive slices of tissue (as measured by overlap in intensity distributions). Moreover, for the majority of antibodies tested, order of use is not critical. For some antibodies fluorescence intensity increases with cycle number and for others it decreases; these factors need to be considered when developing a staining strategy. While the precise reasons for variation in staining with cycle number are not known such variation is reproducible across specimens, suggesting that it reflects properties of the epitope or antibody and not the t-CyCIF process *per se* , variation in staining can be minimized by staining all specimens with the same antibodies in the same order (which also represents the most practical approach). However, this solution is likely to be insufficient for creation of large-scale t-CyCIF datasets in which diverse tissues will be compared with each other (e.g. in proposed tissue atlases [*Department of Health and Human Services, 2018*]) and it will therefore be important to identify antibodies for which cycle number has minimal impact and to create effective methods to correct for those fluctuations that do occur (e.g. inclusion of staining controls).

As an initial application of t-CyCIF, we examined a cancer resection specimen that includes PDAC, healthy pancreas and small intestine. Images were segmented and fluorescence intensities in $\sim 10^5$ whole cells calculated for 24 antibody channels plus a DNA stain. Integrating intensities in this manner does not make use of the many subcellular features visible in t-CyCIF images and therefore represents only a first step in data analysis. We find that expression of vimentin and E-cadherin, classical markers of epithelial and mesenchymal cells, are strongly anti-correlated at a single-cell level and that malignant tissue is skewed toward EMT, consistent with prior knowledge on the biology of pancreatic cancer (*Zeitouni et al., 2016*). The WNT and ERK/MAPK pathways are known to play important roles in the development of PDAC (*Jones et al., 2008*), but the relationship between the two pathways remains controversial. t-CyCIF reveals a negative correlation between β-catenin levels (a measured of WNT pathway activity) and pERK (a measure of MAPK activity) in cells found in some regions of PDAC, non-malignant small intestine and pancreas, a positive correlation in other regions and no significant correlation in yet others. Thus, the full range of discordant observations found in the literature can be recapitulated within a single tumor, emphasizing the wide diversity of signaling states observable at a single-cell level.

As a second application of t-CyCIF, we studied within-tumor heterogeneity in GBM, a brain cancer with multiple histological subtypes whose differing properties impact prognosis and therapy (*Olar and Aldape, 2014*; *Phillips et al., 2006*). Clustering reveals multiple phenotypic classes inter-mingled at multiple spatial scales with no evidence of recurrent patterns. In the GBM we have studied in detail, heterogeneity on a scale of 10–100 cell diameters is as great as it is between distinct lobes. The proportion of cells from different clusters also varies dramatically from one tumor lobe to the next. Although it is not yet possible to link t-CyCIF clusters and known histological subtypes, cell-to-cell heterogeneity on these spatial scales are likely to impact the interpretation of small biopsies (e.g. a core needle biopsy) of a large tumor sample; the data also emphasize the inherent limitation in examining only a small part of a large tumor specimen (e.g. to save time on image acquisition). At the same time, it is important to note that cell-to-cell heterogeneity is caused by processes operating on a variety of time scales, only some of which are likely to be relevant to therapeutic response and disease progression. For example, some cell-to-cell differences visible in GBM images arise from a cyclic process, such as cell cycle progression, whereas others appear to involve differences in cell lineage or clonality. Methods to correct for the effects of variation in cell cycle state have been worked out for single-cell RNA-sequencing (*Izar, 2017*), but will require further work in imaging space.

**Table 3.** Breakdown of individual steps performed for dewaxing and antigen retrieval on a Leica BOND.

| Step | Reagent | Supplier | Incubation (min) | Temp. (°C) |
|---|---|---|---|---|
| 1 | *No Reagent | N/D | 30 | 60 |
| 2 | BOND Dewax Solution | Leica | 0 | 60 |
| 3 | BOND Dewax Solution | Leica | 0 | R.T. |
| 4 | BOND Dewax Solution | Leica | 0 | R.T. |
| 5 | 200 proof ethanol | User* | 0 | R.T. |
| 6 | 200 proof ethanol | User* | 0 | R.T. |
| 7 | 200 proof ethanol | User* | 0 | R.T. |
| 8 | Bond Wash Solution | Leica | 0 | R.T. |
| 9 | Bond Wash Solution | Leica | 0 | R.T. |
| 10 | Bond Wash Solution | Leica | 0 | R.T. |
| 11 | Bond ER1 solution | Leica | 0 | 99 |
| 12 | Bond ER1 solution | Leica | 0 | 99 |
| 13 | Bond ER1 solution | Leica | 20 | 99 |
| 14 | Bond ER1 solution | Leica | 0 | R.T. |
| 15 | Bond Wash Solution | Leica | 0 | R.T. |
| 16 | Bond Wash Solution | Leica | 0 | R.T. |
| 17 | Bond Wash Solution | Leica | 0 | R.T. |
| 18 | Bond Wash Solution | Leica | 0 | R.T. |
| 19 | Bond Wash Solution | Leica | 0 | R.T. |
| 20 | IF Block | User* | 30 | R.T. |
| 21 | Antibody Mix | User* | 60 | R.T. |
| 22 | Bond Wash Solution | Leica | 0 | R.T. |
| 23 | Bond Wash Solution | Leica | 0 | R.T. |
| 24 | Bond Wash Solution | Leica | 0 | R.T. |
| 25 | Hoechst Solution | User* | 30 | R.T. |
| 26 | Bond Wash Solution | Leica | 0 | R.T. |
| 27 | Bond Wash Solution | Leica | 0 | R.T. |
| 28 | Bond Wash Solution | Leica | 0 | R.T. |

DOI: https://doi.org/10.7554/eLife.31657.041

In a third application of t-CyCIF, we characterized tumor-immune cell interactions in a renal cell tumor. Immune checkpoint inhibitors elicit durable responses in a portion of patients with diverse types of cancer, but identifying potential responders and non-responders remains a challenge. In those cancers in which it has been studied (*Mahoney and Atkins, 2014*), quantification of single checkpoint receptors or ligands by IHC lacks sufficient positive and negative predictive value to stratify therapy or justify withholding checkpoint inhibitors in favor of small molecule therapy (*Sharma and Allison, 2015*). Multivariate predictors based on multiple markers such as CD3, CD4, CD8, PD-1 etc. appear to be more effective, but still underperform in patient stratification (*Tumeh et al., 2014*) probably because cells other than CD8 +lymphocytes affect therapeutic responsiveness. In this paper, we perform a simple analysis to show that tumor infiltrating lymphocytes can be subtyped by t-CyCIF and analyzed for the proximity of PD-1 and PD-L1 at a single-cell level. Next steps involve thorough interrogation of immuno-phenotypes by multiplex imaging to relate staining patterns in images to immune cell classes previously defined by flow cytometry and to identify immune cell states that fall below the limit of detection for existing analytical methods.

In conclusion, t-CyCIF is a robust, easy to implement approach to multi-parametric tissue imaging applicable to many types of tumors and tissues; it allows investigators to mix and match antibodies

depending on the requirements of a specific type of sample. To create a widely available community resource, we have posted antibody lists, protocols and example data at http//www.cycif.org and are currently updating this information on a regular basis. Highly multiplexed histology is still in an early stage of development and better methods for segmenting cells, quantifying fluorescence intensities and analyzing the resulting data are in development by multiple groups. The resulting ability to quantify cell-to-cell heterogeneity may enable reconstruction of signaling network topologies in situ (*Giesen et al., 2014*; *Sachs et al., 2002*) by exploiting the fact that protein abundance and states of activity fluctuate from one cell to the next; when fluctuations are well correlated, they are likely to reflect causal associations (*Vilela and Danuser, 2011*). We expect t-CyCIF to be complementary to, and used in parallel with other protein and RNA imaging methods such as FISSEQ (*Lee et al., 2015*) or DEI (*Wang et al., 2017*) that may have higher sensitivity or greater channel capacity. A particularly important task will be cross-referencing tumor cell types identified by single-cell genomics or multi-color flow cytometry with those identified by multiplexed imaging, making it possible to precisely define the genetic geography of human cancer and infiltrating immune cells.

## Competing financial interests

PKS is a member of the Scientific Advisory Board of RareCyte Inc., which manufactures the CyteFinder slide scanner used in this study; research with RareCyte is funded by NIH grant R41 CA224503 (PI E. Kaldjian). PKS is also co-founder of Glencoe Software, which contributes to and supports the open-source OME/OMERO image informatics software used in this paper. Other authors have no competing financial interests to disclose.

# Materials and methods

## Key resources table

| Reagent type (species) or resource | Designation | Source or reference | Identifiers | Additional information |
|---|---|---|---|---|
| Biological sample (human tissue specimen) | TMA:TMA-1207 | Protein Biotechnologies | Cat: TMA-1207 | http://www.proteinbiotechnologies.com/pdf/TMA-1207.pdf |
| Biological sample (human tissue specimen) | TMA:MTU481 | Biomax | Cat: MTU-481 | https://www.biomax.us/tissue-arrays/Multiple_Organ/MTU481 |
| Antibody | Alexa-488 anti-Rabbit antibodies (Fab) | ThermoFisher Scientific | Cat: A-11034 (RRID:AB_2576217) | Dilution 1:2000 |
| Antibody | Alexa-555 anti-Rat antibodies | ThermoFisher Scientific | Cat: A-21434 (RRID:AB_141733) | Dilution 1:2000 |
| Antibody | Alexa-647 anti-Mouse antibodies (Fab) | ThermoFisher Scientific | Cat: A-21236 (RRID:AB_141725) | Dilution 1:2000 |
| Chemical compound, drug | Hoechst 33342 | ThermoFisher Scientific | Cat: H3570 | https://www.thermofisher.com/order/catalog/product/H3570 |
| Software, algorithm | ImageJ | PMID:22930834 | RRID: SCR_003070 | https://imagej.nih.gov/ij/ |
| Software, algorithm | Matlab | MathWorks, Inc. | RRID:SCR_001622 | |
| Software, algorithm | Ashlar | Laboratory of Systems Pharmacology, Harvard Medical School | RRID:SCR_016266 | https://github.com/sorgerlab/ashlar (copy archived at https://github.com/elifesciences-publications/ashlar) |
| Software, algorithm | BaSiC | Helmholtz Zentrum München | RRID: SCR_016371 | https://www.nature.com/articles/ncomms14836 |
| Other | www.cycif.org | Laboratory of Systems Pharmacology, Harvard Medical School | RRID:SCR_016267 | Online resource for cyclic immunofluorescence |
| Other | lincs.hms.harvard.edu | HMS LINCS Center | RRID:SCR_016370 | Additional data/image resource for t-CyCIF |

Key resources, reagents and software used in this study are listed in Key resources table and also online at the HMS LINCS Center Publication Page http://lincs.hms.harvard.edu/lin-elife-2018/ (RRID: SCR_016370). This page provides links to an OMERO image database from which individual images can be obtained; stitched and registered image panels can be obtained at www.cycif.org (RRID: SCR_016267) and a video illustrating the t-CyCIF method can be found at https://vimeo.com/269885646. The data on staining repeatability shown in *Figures 5* and *6* are complex and are available in a Jupyter notebook at https://github.com/sorgerlab/lin_elife_2018_tCyCIF_plots (*Muhlich and Wang, 2018*; copy archived at https://github.com/elifesciences-publications/lin_elife_2018_tCyCIF_plots).

## Patients and specimens

Formalin fixed and paraffin embedded (FFPE) tissues from were retrieved from the archives of the Brigham and Women's Hospital as part of discarded/excess tissue protocols or obtained from commercial vendors. The Institutional Review Board (IRB) of the Harvard Faculty of Medicine last reviewed the research described in this paper on 2/16/2018 (under IRB17-1688) and judged it to 'involve no more than minimal risk to the subjects' and thus eligible for a waiver of the requirement to obtain consent as set out in 45CFR46.116(d).

Tumor tissue and FFPE specimens were collected from patients under IRB-approved protocols (DFCI 11–104) at Dana-Farber Cancer Institute/Brigham and Women's Hospital, Boston, Massachusetts. Tonsil samples used in *Figure 1* were purchased from American MasterTech (CST0224P). Tissue microarrays for analyses in *Figure 4D and E* were obtained from Biomax (Cat. MTU481); detailed information can be found online at https://www.biomax.us/tissue-arrays/Multiple_Organ/MTU481. Tissue microarrays (TMA) for diverse healthy tissues and tumor analyses were obtained from Protein Biotechnologies (Cat. TMA-1207).

## Reagents and antibodies

All conjugated and unconjugated primary antibodies used in this study are listed in *Table 2*. Indirect immunofluorescence was performed using secondary antibodies conjugated with Alexa-647 anti-Mouse (Invitrogen, Cat. A-21236), Alexa-555 anti-Rat (Invitrogen, Cat. A-21434) and Alexa-488 anti-Rabbit (Invitrogen, Cat. A-11034). 10 mg/ml Hoechst 33342 stock solution was purchased from Life Technologies (Cat. H3570). 20xPBS was purchased from Santa Cruz Biotechnology (Cat. SC-362299). 30% hydrogen peroxide solution was purchased from Sigma-Aldrich (Cat. 216763). PBS-based Odyssey blocking buffer was purchased from LI-COR (Cat. 927–40150). All reagents for the Leica BOND RX were purchased from Leica Microsystems. HCS CellMask Red Stain and Mito-tracker Green stains were purchased from ThermoFischer (catalog numbers H32712, R37112 and M751, respectively).

## Pre-processing and pre-staining tissues for t-CyCIF

### Automated dewaxing, rehydration and pre-staining

Pre-processing of FFPE tissue and tumor slices mounted on slides was performed on a Leica BOND RX automated stained using the protocol shown in *Table 3*.

Steps 2–10: Dewaxing and Rehydration with Leica Bond Dewax Solution Cat. AR9222.

Steps 11–14: Antigen retrieval with BOND Epitope Retrieval solution 1 (ER1; Cat. AR9961).

Steps 15–19: Washing with Leica Bond Wash Solution (Cat. AR9590).

Steps 20–28 Pre-staining procedures as shown in *Figure 1A*:

Step 20: IF Block - Immunofluorescence blocking in Odyssey blocking buffer (LI-COR, Cat. 927401).

Step 21: Antibody Mix - Incubation with secondary antibodies diluted in Odyssey blocking buffer.

Step 25: Staining with Hoechst 33342 at 2 µg/ml (w/v) in in Odyssey blocking buffer.

### Manual dewaxing, rehydration and pre-staining

In our experience dewaxing, rehydration and pre-staining can also be performed manually with similar results. For manual pre-processing, FFPE slides were first incubated in a 60°C oven for 30 min. To completely remove paraffin, slides were placed in a glass slide rack and then immediately immersed in Xylene in a glass staining dish (Wheaton 900200) for 5 min and subsequently transferred to another dish containing fresh Xylene for 5 min. Rehydration was achieved by sequentially immersing

slides, for 3 min each, in staining dishes containing 100% ethanol, 90% ethanol, 70% ethanol, 50% ethanol, 30% ethanol, and then in two successive 1xPBS solutions. Following rehydration, slides were placed in a 1000 ml beaker filled with 500 ml citric acid, pH 6.0, for antigen retrieval. The beaker containing slides and citric acid buffer was microwaved at low power until the solution was at a boiling point and maintained at that temperature for 10 min. After cooling to room temperature, slides were washed 3 times with 1xPBS in vertical staining jars.

### Prestaining

Dewaxed specimens were blocked by incubation with Odyssey blocking buffer for 30 mins by applying the buffer to slides as a 250–500 µl droplet at room temperature; evaporation was minimized by using a slide moisture chamber (Scientific Device Laboratory, 197-BL). Slides were then pre-stained by incubation with diluted secondary antibodies (listed above) for 60 min, followed by washing three times with 1xPBS. Finally, slides were incubated with Hoechst 33342 (2 µg/ml) in 250–500 µl Odyssey blocking buffer for 30 min in a moisture chamber and washed three times with 1xPBS in vertical staining jars. After imaging, cells were subjected to a round of fluorophore inactivation (see below). Following fluorophore inactivation, slides were washed four times with 1x PBS by dipping them in a series of vertical staining jars to remove residual inactivation solution.

## Performing cyclic immunofluorescence

All primary antibodies (fluorophore-conjugated and unconjugated) were diluted in Odyssey blocking buffer. Slides carrying tissues that had been subjected to pre-staining, or to a previous t-CyCIF stain and bleach cycle, were incubated at 4°C for ~12 hr with diluted primary or fluorophore-conjugated antibody (250–500 µl per slide) in a moisture chamber. Long incubation times were a matter of convenience and many antibodies only require short incubation with sample. Slides were then washed four times in 1x PBS by dipping in a series of vertical staining jars.

For indirect immunofluorescence, slides were incubated in diluted secondary antibodies in a moisture chamber for 1 hr at room temperature followed by four washes with 1xPBS. Slides were incubated in Hoechst 33342 at 2 µg/ml in Odyssey blocking buffer for 15 min at room temperature, followed by four washes in 1xPBS. Stained slides were mounted prior to image acquisition (see the Mounting section below).

### Primary antibodies

For t-CyCIF, we selected commercial antibodies previously validated by their manufacturers for use in immunofluorescence, immunocytochemistry or immunohistochemistry (IF, ICC or IHC). When possible, we checked antibodies on reference tissue known to express the target antigen, such as immune cells in tonsil tissue or tumor-specific markers in tissue microarrays. The staining patterns for antibodies with favorable signal-to-noise ratios were compared to those previously reported for that antigen by conventional antibodies. An updated list of all antibodies tested to date can be found at http://www.cycif.org. In current practice, the degree of validation is quantified on a level between 0 and 2: 'Level 0' represents antibodies with inconsistent or no staining in tissues for which the antigen is thought to be present based on published data; 'Level 1' represents the expected pattern of positive staining in a limited number of tissues types (e.g. CD4 antibody in tonsil tissue alone); 'Level 2' represents the expected pattern of positive staining in all tissues or tumor types tested (N >= 3). Higher levels will be assigned in the future to antibodies that have undergone extensive validation; for example, side-by-side comparison of against an established IHC positive control. Overall, the validation of primary antibodies used in this study is not meaningfully greater what has already been done by commercial vendors using conventional IF or IHC.

### Mounting and de-coverslipping

Immediately prior to imaging, slides were mounted with 1xPBS or, if imaging was expected to take longer than 30 min, for example, in the case of samples larger than 2–4 cm$^2$ (corresponding to about 200 fields of view with a 10X objective) PBS was supplement with 10% Glycerol. Slides were covered using 24 × 60 mm No. one coverslips (VWR 48393–106) to prevent evaporation while facilitating subsequent de-coverslipping via gravity. Following image acquisition, slides were placed in a vertical

staining jar containing 1xPBS for at least 15 min. Coverslips were released from slides (and the tissue sample) via gravity as the slides were slowly drawn out of the staining jar.

## Fluorophore inactivation (bleaching)

After imaging, fluorophores were inactivated by placing slides horizontally in 4.5% $H_2O_2$ and 24 mM NaOH made up in PBS for 1 hr at RT in the presence of white light. Following fluorophore inactivation, slides were washed four times with 1x PBS by dipping them in a series of vertical staining jars to remove residual inactivation solution.

## Image acquisition

Stained slides from each round of CyCIF were imaged with a CyteFinder slide scanning fluorescence microscope (RareCyte Inc. Seattle WA) using either a 10X (NA = 0.3) or 40X long-working distance objective (NA = 0.6). Imager5 software (RareCyte Inc.) was used to sequentially scan the region of interest in four fluorescence channels. These channels are referred to by the manufacturer as a: (i) 'DAPI channel' with an excitation filter having a peak of 390 nm and half-width of 18 nm and an emission filter with a peak of 435 nm and half-width of 48 nm; (ii) 'FITC channel' having a 475/28 nm excitation filter and 525/48 nm emission filter (iii); 'Cy3 channel' having a 542/27 nm excitation filter and 597/45 nm emission filter and (iv); 'Cy5 channel' having a 632/22 nm excitation filter and 679/34 nm emission filter. Imaging was performed with $2 \times 2$ binning to increase sensitivity, shorten exposure time and reduce photo bleaching. We have tested slide scanners from several other manufacturers (e.g. a Leica Aperio Digital Pathology Slide Scanner, GE IN-Cell Analyzer 6000 and GE Cytell Cell Imaging System) and found that they too can be used to acquire images from samples processed by t-CyCIF. Slides can also be analyzed on conventional microscopes, but the field of view is typically smaller, and an automated stage is required for accurate stitching of individual fields of view into a complete image of a tissue.

## Super-resolution microscopy

We acquired 3D-SIM images on a Deltavision OMX V4 Blaze (GE Healthcare) with a 60x/1.42N.A. Plan Apo oil immersion objective lens (Olympus) and three Edge 5.5 sCMOS cameras (PCO). Two to three micron z-stacks were collected with a z-step of 125 nm or 250 nm and with 15 raw images per plane. To minimize spherical aberration, immersion oil matching was used for each sample as described by *Hiraoka et al. (1990)*. except that we measured point spread functions of point-like structures within the sample as opposed to beads on a separate slide. DAPI fluorescence was excited with a 405 nm laser and collected with a 477/35 emission filter, Alexafluor 488 with a 488 nm laser and a 528/48 emission filter, Alexa fluor 555 with a 568 nm laser and a 609/37 emission filter, and Alexa fluor 647with a 642 nm laser and a 683/40 emission filter. All stage positions were saved in softWorX to be revisited later. Super-resolution images were computationally reconstructed from the raw data sets with a channel-specific, measured optical transfer function and a Wiener filter constant of 0.001 using CUDA-accelerated 3D-SIM reconstruction code based on *Gustafsson et al. (2008)*. A comparison of properties of different imaging platforms used in this study are shown in *Table 1*.

## Image processing

Quantitative analysis of tissue images is challenging, in large part because cells are close together and embedded in a complex extracellular environment. Background can be uneven across large images and signal-to-noise ratios relatively low, particularly in the case of tissues with high auto-fluorescence and low signal antibodies (e.g. phospho-protein antibodies). We have only started to tackle these issues in the case of high-dimensional t-CyCIF data and users are encouraged to check for updates on www.cycif.org and implement their own approaches.

## Background subtraction and image registration

Background subtraction was performed using the previously established rolling ball algorithm (with a 50-pixel radius) in ImageJ. Adjacent background-subtracted images from the same sample were then registered to each using an ImageJ script as described previously (*Lin et al., 2015*). All images with 2×2 binning in acquisition were partially de-convoluted with unsharp masking. DAPI images

from each cycle were used to generate reference coordinates by Rigid-body transformation. To generate virtual hyper-stacked images, the transformed coordinates were applied to images from four channel imaging of each t-CyCIF cycle.

## Single-cell segmentation and quantification

To obtain intensity values for single cells, images were segmented using a previously described (*Lin et al., 2015*) Watershed algorithm based on nuclear staining by Hoechst 33342. Images were initially thresholded using the OTSU algorithm and binarized in the Hoechst channel, which was then used to generate a nuclear mask image. The mask images were then subjected to the Watershed algorithm in ImageJ to obtain single-cell regions of interest (ROIs). From the nuclei, the cytoplasm was captured by centripetal expansion of either of 3 pixels in images obtained with a 10X objective or of 6 pixels in images obtained with a 40X objective, until cell reaching the cell boundaries (cell membrane). The cytoplasm was then defined as the region between the cell membrane and the nucleus. Following cell segmentation, these cell boundaries were used to compute mean and integrated intensity values from all channels. Because ROIs are (initially) defined only by the nuclear signal, this approach is likely to over- or under- segment cells with irregular shapes, which can lead to nuclear, cytosolic or cell membrane 'signal contamination' between neighboring and/or stacked cells. Further experimental (e.g. including membrane markers to guide whole-cell rather than nuclear-only segmentation) and analytical algorithms to more accurately segment individual cells (e.g. using deep learning methods to register and apply additional features) would help to improve segmentation. All imageJ scripts used in this manuscript can be found in our Github repository (https://github.com/sorgerlab/cycif [*Lin, 2018*]; copy archived at https://github.com/elifesciences-publications/cycif).

## Image stitching, shading and flat-field correlation

The BaSiC algorithm (*Peng et al., 2017*) was used for shade and flat-field correction in the create of the multi-panel montage images shown in *Figures 2B*, *6B*, *9A* and *11A*. Additional information can be found on the BaSiC website (https://www.helmholtz-muenchen.de/icb/research/groups/quantitative-single-cell-dynamics/software/basic/index.html). An example of the performance of BaSiC is shown in *Figure 2—figure supplement 1*. The ImageJ plugin of BaSiC was applied for whole image stacks using the default options. After processing with BaSiC, images stack were stitched with ImageJ/Fiji 'Grid stitch' plugin with default options. ASHLAR was used to stich, register and scale images available at http://www.cycif.org/.

## Time considerations

We believe that the greater time invested in t-CyCIF as compared to conventional IF IHC must be placed in the context of the much greater amount of data generate from a t-CyCIF experiment. It is also important to note that while t-CyCIF can be relatively slow when a single sample is processed it can easily be performed in parallel on multiple samples. As a practical example, we usually stain 30 slides in parallel (each involving 100-200 fields of view); in the case of TMAs, >80 samples can be assembled on each slide, so up to 2400 samples can be processed in parallel. With a single scanner, 30 slides can be scanned (average scan time ~10 min) in about 6 hr. Photo-inactivation and washing steps take ~1 to 1.5 hr, after which an additional round of staining is initiated. As a matter of convenience, we usually perform staining overnight. Hence, one user can generate data for 90 channels and 1800 images per day. Thus, ~10 work days are required to generate 900 channels/18,000 images. Further time needs to be allotted for registration and stitching (~12–18 hr of computing time) and quantification (~24–48 hr computing time, depending on cell density). Overall, we believe that this is a reasonable level of throughput; moreover we have not yet attempted to optimize it using fluidic devices, automated stainers etc. We also note that the throughput of t-CyCIF compares favorably with other tissue-imaging platforms and single-cell transcriptome profiling.

## Analysis of tissue integrity over cycles

We purchased a TMA (MTU481, Biomax Inc, https://www.biomax.us/tissue-arrays/Multiple_Organ/MTU481) to test the impact of cycle number on tissue integrity. Images were captured and processed as described above. The registered image stacks were then segmented and nuclei counts for

each core and each cycle were recorded. All values were normalized to the number of nuclei from the first cycle of a particular core biopsy and the fractional normalized nuclei count shown at each staining cycle.

## Calculation of intensity overlap between different cycles and dynamic range

To compare staining patterns between different cycles within the same specimen, we calculated overlap integrals. First, we determined the distribution of intensity data averaged over each single cell and for each t-CyCIF cycles. The area under the curve of these distributions was calculated by trapezoidal numerical integration using 'trapz' function in Matlab (*Gustafsson et al., 2008*). The ratio of the area under the curve (AUC) for different cycles, samples or antibodies was calculated and the overlap scores then computed as:

$$Overlap\,score = overlap\,AUC/total\,AUC$$

The dynamic range (DR) of fluorescence intensities for a given antibody was calculated as a rough estimate of the signal-to-noise ratio; SNR. The calculation was performed as follows: first, pixel-by-pixel intensity data was extracted from a t-CyCIF image; the DR was then calculated as the ratio of the intensities of the 95th and 5th percentile values and represented on a log scale. High DR values indicate a favorable SNR. Intensities below the 5th percentile were considered to be background noise.

## High-dimensional single-cell analysis by t-SNE

Raw intensity data generated from registered and segmented images were imported into Matlab and converted to comma separated value (csv) files. The viSNE implementation of t-SNE and EMGM algorithms from the CYT single-cell analysis package were obtained from the Pe'er laboratory at Columbia University (*Amir et al., 2013*). Intensity-based measurements (such as flow cytometry or imaging cytometry) of protein expression have approximately log-normal distribution (*Bagwell, 2005*), hence, t-CyCIF raw intensity values were first transformed in log or in inverse hyperbolic sine (*asinh*) using the default Matlab function or the *CYT* package (*Amir et al., 2013*), respectively. Between-sample variation was normalized on a per-channel basis by using the *CYT* package to align intensity measurements that encompass values between 1st and the 99th percentile. Data files were aggregated and used to generate viSNE plots. All viSNE/t-SNE analyses used the following settings: perplexity −30, epsilon = 500, lie factor = 4 for initial 100 iterations and lie factor −1 for remaining iterations.

## Regional and neighboring analysis using K-nearest neighbors (KNN) methods

To determine whether PD-1 and PD-L1 expressing cells are sufficiently close for the receptor and ligand to interact, the spatial densities for PD1$^+$ and PDL1$^+$ cells were estimated using a k nearest neighbors (kNN) model with k = 4, corresponding to a ~10 μm smoothing window. Since the density in space of the PD1$^+$ or PDL1$^+$ cells at any point in that space is proportional to the probability of that cell having a centroid there, the co-occurrence probability at a point was therefore proportional to the product of the spatial densities for both cell types at a point. To normalize for the difference in total PDL1+ or PD1+ cells between regions of the tissue corresponding to tumor and stroma, we calculated spatial probabilities for the different regions in the specimen separately. *Figure 9—figure supplement 1* shows the distribution of co-occurrence densities for stroma and tumor relevant to a clear-cell carcinoma shown in *Figure 9*.

## Calculating Shannon entropy values

Images were divided into regular grids and 1000 cells from each region used to calculate the non-parametric Shannon entropy as follows:

$$Shanon\,Entropy\,(s) = -i\sum s_i^2 \log\left(s_i^2\right)$$

where $s_i$ is the per-pixel intensity of signal **s** at a given point. Normalized Shannon entropy as calculated as $E_{normalized} = E_{region}/E_{sample}$.

## Expectation–Maximization Gaussian mixtures (EMGM) clustering

To determine an appropriate number of clusters ($k$) for analysis of the GBM tumor shown in **Figures 11** and **12** and in **Figure 12—figure supplement 2** we determined negative log-likelihood-ratios for various values of $k$. For each choice of cluster number $n$, the likelihood-ratio was calculated for a Gaussian mixture model with $n = k-1$ and with $n = k$ and the ratio then plotted relative to k. The EMGM algorithm was initialized 30 times for each value of $k$ and it converged in all instances. The inflection at k = 8 (red arrow) suggests that inclusion of additional clusters (k > 8) explains a smaller, distinct source of variation in the data (**Figure 12—figure supplement 1**). As an alternative, k = 12 was also explored in **Figure 12—figure supplement 2**. Intensity values from all antibody channels (plus area and Hoechst intensity) were used for clustering.

## Data availability

All data generated or analyzed during this study are included in the manuscript and supporting files. Intensity data used to generate figures is available in supplementary materials and can be downloaded from the HMS LINCS Center Publication Page (http://lincs.hms.harvard.edu/lin-elife-2018/) (RRID:SCR_016370).

## Code availability

Code and scripts used in this study are listed in Key resources table and also on-line at the HMS LINCS Center publication page (http://lincs.hms.harvard.edu/lin-elife-2018/). ImageJ is available at https://imagej.nih.gov/ij/

BaSic is available at https://www.helmholtz-muenchen.de/icb/research/groups/quantitative-single-cell-dynamics/software/basic/index.html. Matlab scripts used in this paper and the ASHLAR registration/stitching algorithm is available at our GitHub repositories (https://github.com/sorgerlab/cycif and https://github.com/sorgerlab/ashlar (**Muhlich, 2018**; **Lin, 2018**). A Jupyter notebook for futher exploration of data in **Figures 5** and **6** is available at https://github.com/sorgerlab/lin_elife_2018_tCyCIF_plots (**Muhlich and Wang, 2018**; copy archived at https://github.com/elifesciences-publications/lin_elife_2018_tCyCIF_plots).

## Image availability

All images can be obtained from an OMERO image database via links found at the HMS LINCS Center Publication Page http://lincs.hms.harvard.edu/lin-elife-2018/ (RRID: SCR_016370). Stitched and registered image composites can be obtained at www.cycif.org. (RRID:SCR_016267) and via links found there.

## Acknowledgements

This work was funded by NIG grants P50-GM107618 (PKS), U54-HL127365 (PKS), and R41-CA224503 (PKS) and by a DF/HCC GI SPORE Developmental Research Project Award (BI) and DFCI Claudia Adams Barr Program for Innovative Cancer Research Award (BI). BI was also supported by grant K08CA222663. SW was also supported by NIH/NIGMS training grant T32GM008313. We thank J Waters and T Lambert from the Harvard Cell Biology Microscopy Facility for access to the OMX Blaze, their guidance on SIM acquisition and analysis, and L Shao for CUDA-accelerated SIM reconstruction code., B Wolpin and C Lian for providing specimens, Z Maliga and J Muhlich for technical support and L Garraway, and members of Ludwig Center for Cancer Research at Harvard for many fruitful discussions.

## Additional information

### Competing interests
Peter K Sorger: PKS is a member of the Board of Directors of RareCyte Inc., which manufactures the slide scanner used in this study, and co-founder of Glencoe Software, which contributes to and supports open-source OME/OMERO image informatics software. Other authors have no competing financial interests to disclose. The other authors declare that no competing interests exist.

### Funding

| Funder | Grant reference number | Author |
| --- | --- | --- |
| National Institutes of Health | P50GM107618 | Peter K Sorger |
| Dana-Farber/Harvard Cancer Center | GI SPORE Developmental Research Project Award | Benjamin Izar |
| National Institutes of Health | U54HL127365 | Peter K Sorger |
| National Institutes of Health | R41-CA224503 | Peter K Sorger |
| Dana-Farber/Harvard Cancer Center | Claudia Adams Barr Program | Benjamin Izar |
| National Institutes of Health | K08CA222663 | Benjamin Izar |

The funders had no role in study design, data collection and interpretation, or the decision to submit the work for publication.

### Author contributions
Jia-Ren Lin, Conceptualization, Data curation, Software, Formal analysis, Validation, Investigation, Visualization, Methodology, Writing—original draft, Writing—review and editing; Benjamin Izar, Conceptualization, Resources, Data curation, Supervision, Funding acquisition, Validation, Investigation, Writing—original draft, Writing—review and editing; Shu Wang, Data curation, Formal analysis, Writing—review and editing; Clarence Yapp, Investigation, Methodology, Writing—review and editing; Shaolin Mei, Resources, Data curation, Formal analysis, Investigation; Parin M Shah, Resources, Data curation; Sandro Santagata, Data curation, Supervision, Project administration, Writing—review and editing; Peter K Sorger, Conceptualization, Supervision, Funding acquisition, Investigation, Methodology, Writing—original draft, Project administration, Writing—review and editing

### Author ORCIDs
Jia-Ren Lin (iD) http://orcid.org/0000-0003-4702-7705
Benjamin Izar (iD) http://orcid.org/0000-0003-2379-6702
Peter K Sorger (iD) http://orcid.org/0000-0002-3364-1838

### Ethics
Human subjects: Formalin fixed and paraffin embedded (FFPE) tissues were retrieved from the archives of the Brigham and Women's Hospital as part of discarded/excess tissue protocols or obtained from commercial vendors. The Institutional Review Board (IRB) of the Harvard Faculty of Medicine last reviewed the research described in this paper on 2/16/2018 (under IRB17-1688) and judged it to 'involve no more than minimal risk to the subjects' and thus eligible for a waiver of the requirement to obtain consent as set out in 45CFR46.116(d). Tumor tissue and FFPE specimens were collected from patients under IRB-approved protocols (DFCI 11-104) at Dana-Farber Cancer Institute/Brigham and Women's Hospital, Boston, Massachusetts. The consent waiver described above also covers these tissues and specimens.

### Decision letter and Author response
Decision letter https://doi.org/10.7554/eLife.31657.050
Author response https://doi.org/10.7554/eLife.31657.051

# Additional files

## Supplementary files

• Supplementary file 1. List of antibodies used for staining in *Figure 3*.
DOI: https://doi.org/10.7554/eLife.31657.042

• Supplementary file 2. List of antibodies used for staining in *Figures 5* and *6*.
DOI: https://doi.org/10.7554/eLife.31657.043

• Supplementary file 3. List of antibodies used for staining in *Figures 7*, *8* and *10*.
DOI: https://doi.org/10.7554/eLife.31657.044

• Supplementary file 4. List of antibodies used for staining in *Figure 9*.
DOI: https://doi.org/10.7554/eLife.31657.045

• Supplementary file 5. Descriptions of TMA shown in *Figure 10*.
DOI: https://doi.org/10.7554/eLife.31657.046

• Supplementary file 6. List of antibodies used for staining in *Figures 11* and *12*.
DOI: https://doi.org/10.7554/eLife.31657.047

• Transparent reporting form
DOI: https://doi.org/10.7554/eLife.31657.048

## Data availability

All data generated or analyzed during this study are included in the manuscript and supporting files. Intensity data used to generate figures is available in supplementary materials and can be downloaded from the HMS LINCS Center Publication Page (http://lincs.hms.harvard.edu/lin-elife-2018/) (RRID:SCR_016370). The images described are available at http://www.cycif.org/ (RRID:SCR_016267) and via and OMERO server as described at the LINCS Publication Page.

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
