## [Decision Letter]

Thank you for submitting your article "A simple open-source method for highly multiplexed imaging of single cells in tissues and tumours" for consideration by *eLife*. Your article has been reviewed by Arup Chakraborty as the Senior Editor, a Reviewing Editor, and three reviewers. The following individuals involved in review of your submission have agreed to reveal their identity: Carsten Marr (Reviewer #2); Péter Horváth (Reviewer #3).

The reviewers have discussed the reviews with one another and the Reviewing Editor has drafted this decision to help you prepare a revised submission.

Summary:

Overall, the reviewers appreciated the ability to multiplex immunofluorescence in FFPE samples using cyclic chemistry to measure expression and localization of several proteins in single cells. The open-source tools for the image processing pipeline were also thought to be of much interest to the community.

Essential revisions:

As discussed in our earlier correspondence, one concern that arose during deliberations was the lack of a clear discussion of the advance described in this manuscript relative to other contributions in the field, in particular that of Gerdes et al. We appreciate the arguments about lack of adoption of the method described by Gerdes et al., and its proprietary nature, and ultimately believe that for methodological improvements like this, the research community may be the best judge of the relative merits. Nevertheless, we do think it's very important to clearly delineate the contributions of the previous work in the field and precisely what the advance is in this present manuscript relative to those contributions, both in the Introduction and Discussion section.

Technically, the reviewers felt that the work lacked sufficient testing to show that order of antigen staining is not affected by cycle number, noting "This is a well known issue for multiplexed tissue staining and should be analyzed beyond just 4 cycles with 3 antigens. Tissue integrity is compromised after 5 cycles and only quantified to 10 cycles (Figure 1H); the authors claim their methods work to 20 cycles but descriptions of tissue integrity are lacking." The recommendation was: "Retention of antigenicity is only showed up to cycle 4 and for only 3 antigens, although data from higher cycles are used in other figures (20 cycles in Figure S3). Gerdes et al. demonstrates that 8/59 antibodies tested did not maintain full antigenicity after the tissue had been exposed 10 times to dye inactivation solution. We recognize that an experiment to test all possible orders/combinations of antibodies would be time and labor intensive, but we believe antibody validation must include how long antigenicity is preserved through cycles. We suggest the following tests of the methods:

* Two adjacent tissue slices are stained for 20 cycles with antibodies 1->20 and 20->1, respectively. The results are shown to be at least qualitatively similar.

* Maintenance of antigenicity for most, if not all, single antibodies up to 10-20 cycles."

Another reviewer noted "As a quantitative method, I would appreciate an evaluation of the robustness of the single cell measurements over cycles. It would be interesting to see how single cell intensities correlate when stained for the same antigens in cycle 1, cycle 2, cycle 3 etc., maybe even using different fluorophores, or a staining in cycle 1 and again in cycle 10, with other antigen stainings in between. This would add a quantitative level to Figure 2B."

Also: "Would be great to know the authors' experiences regarding the degradation after 8-20 t-CyCIF cycles, which is only partially discussed. For basic biology discovery studies, it would be great to have a stopping criteria where the number of washing steps saturate and noise takes over the signal, and in potential clinical practice a cycle number until quality is guaranteed would also be desired."

We feel that these technical points are important to fully address in a revision.

---

## [Author Response]

Summary:Overall, the reviewers appreciated the ability to multiplex immunofluorescence in FFPE samples using cyclic chemistry to measure expression and localization of several proteins in single cells. The open-source tools for the image processing pipeline were also thought to be of much interest to the community.Essential revisions:As discussed in our earlier correspondence, one concern that arose during deliberations was the lack of a clear discussion of the advance described in this manuscript relative to other contributions in the field, in particular that of Gerdes et al. We appreciate the arguments about lack of adoption of the method described by Gerdes et al., and its proprietary nature, and ultimately believe that for methodological improvements like this, the research community may be the best judge of the relative merits. Nevertheless, we do think it's very important to clearly delineate the contributions of the previous work in the field and precisely what the advance is in this present manuscript relative to those contributions, both in the Introduction and Discussion section.

We certainly agree with the reviewer that prior studies should be adequately cited; we had not intended to slight prior work by Gerdes and others (although we agree that we did not do a good job in the first submission). To address this concern we have re-written the Introduction and Discussion section to specifically mention prior work by Gerdes and to make clear that our paper builds on that earlier work.

Technically, the reviewers felt that the work lacked sufficient testing to show that order of antigen staining is not affected by cycle number, noting "This is a well known issue for multiplexed tissue staining and should be analyzed beyond just 4 cycles with 3 antigens. Tissue integrity is compromised after 5 cycles and only quantified to 10 cycles (Figure 1H); the authors claim their methods work to 20 cycles but descriptions of tissue integrity are lacking." The recommendation was: "Retention of antigenicity is only showed up to cycle 4 and for only 3 antigens, although data from higher cycles are used in other figures (20 cycles in Figure S3). Gerdes et al. demonstrates that 8/59 antibodies tested did not maintain full antigenicity after the tissue had been exposed 10 times to dye inactivation solution. We recognize that an experiment to test all possible orders/combinations of antibodies would be time and labor intensive, but we believe antibody validation must include how long antigenicity is preserved through cycles. We suggest the following tests of the methods:*Two adjacent tissue slices are stained for 20 cycles with antibodies 1->20 and 20->1, respectively. The results are shown to be at least qualitatively similar.*Maintenance of antigenicity for most, if not all, single antibodies up to 10-20 cycles."Another reviewer noted "As a quantitative method, I would appreciate an evaluation of the robustness of the single cell measurements over cycles. It would be interesting to see how single cell intensities correlate when stained for the same antigens in cycle 1, cycle 2, cycle 3 etc., maybe even using different fluorophores, or a staining in cycle 1 and again in cycle 10, with other antigen stainings in between. This would add a quantitative level to Figure 2B."

These are all very important issues and we have spent the extended revision period performing multiple experiments to address them. Three separate 16-cycle antibody swap experiments, each involving two immediately adjacent tissue slides, were performed to study the issue of antibody order of addition. In these studies antibodies against abundant proteins were applied repeatedly to successive specimens from the same tissue block (slides A and B). Abundant proteins are expected to be relatively unaffected by antibody saturation and were used in four cycles spread across 16 cycles total. The impact of cycle number on antibodies against less abundant antigens (which are potentially easier to saturate) was evaluated by swapping them between early and late cycles on slides A and B. This made it possible to assess several issues raised by the reviewers, including (1) the repeatability of staining on a single sample, (2) the reproducibility of staining across specimens, and (3) the effect of swapping between early and late cycles on morphology and fluorescence intensity. We also examined signal to noise ratio and tissue integrity more carefully than previously.

Results from these studies are presented in two new figures (Figure 5, Figure 6 and Figure 6—figure supplement 1) and in a new section of the Results section. Overall, we find little or no evidence that antigenicity falls across the board as cycle number increases. We have confirmed that signal-to-noise ratios can increase with higher cycle number due to lower auto-fluorescence. For a subset of antibodies, we do observe significant changes in fluorescence intensity with cycle number but these can involve both increases and decreases in intensity. When antibodies are used in the same cycle across two samples, a very high degree of repeatability is possible (Figure 6F).

Tissue integrity and not antigenicity appears to be the primary limitation on high-cycle t-CyCIF. We find that about half of all tissue tested can routinely be imaged out to 15 cycles with <20% loss of cells but that other tissues are less robust. Considerable variability is observed within a single tissue type (data on breast cancer is shown in Figure 4E). In response to the reviewer’s concerns we have toned down our claim about “60 channels and 20 cycles” although we now show such an experiment in its entirety in Figure 4F-G and we continue to perform high-cycle t-CyCIF on a routine basis. We speculate that “pre-analytical variables” such fixation conditions, the age of tissue blocks and similar variables strongly influence tissue integrity; we are studying this now and expect to return to it in a future paper.

One additional issue affecting cycle-to-cycle reproducibility is the lack of sensors, in current slide scanners, to measure and adjust the intensity of excitatory illumination. These instruments do not illuminate the back focal plane with the uniformity expected of high resolution microscopes. Because of stage instability it is also difficult to ensure that repeated sampling occurs at the same position in Z. All of these issues can be addressed with additional hardware development and we are fortunate to have recently obtained an NIH STTR grant to co-fund hardware development with RareCyte, manufacturer of the instruments used in this study. We touch very briefly on these issues in the revised Discussion section.

Also: "Would be great to know the authors' experiences regarding the degradation after 8-20 t-CyCIF cycles, which is only partially discussed. For basic biology discovery studies, it would be great to have a stopping criteria where the number of washing steps saturate and noise takes over the signal, and in potential clinical practice a cycle number until quality is guaranteed would also be desired."

We appreciate the reviewer’s concern: knowing how many cycles can be performed on different types of tissue will be important. The revision now describes a 10-cycle t-CyCIF experiment on a tissue microarray (TMA) comprising 48 core biopsies derived from 16 different healthy tissue types and several breast cancers (Figure 4). To measure tissue integrity, we quantified the number of nuclei and plotted the normalized nuclei count (relative to the pre-staining nuclei count) after each staining cycle (Figure 4D). All samples could undergo 10 cycles of staining with retention of >70% or cells but integrity varied between different tissue types and across biopsies from the same tumor type (Figure 4E).

We conclude that tissues can reliably be subjected to 8 to 10-cycle t-CyCIF and some specimens maintain their integrity even after 20 cycles. Unfortunately, we do not yet understand the factors underlying differences in tissue integrity except that they are likely to involve biological factors, preanalytical variables (i.e. fixation, age of tissue blocks, mounting, and cutting) and the t-CyCIF process itself. Fortunately, it is very easy to measure tissue integrity empirically after each cycle based on the fraction of nuclei still present in the sample and users of the method need to do this themselves. Future work will be required to develop useful predictors and systematic improvements in cycle number.